# Identifiability for Gaussian Processes with Holomorphic Kernels

**Ameer Qaqish,   Didong Li**
Department of Biostatistics, University of North Carolina at Chapel Hill
{ameer,didongli}@unc.edu

## Abstract

Gaussian processes (GPs) are widely recognized for their robustness and flexibility across various domains, including machine learning, analysis of time series, spatial statistics, and biomedicine. In addition to their common usage in regression tasks, GP kernel parameters are frequently interpreted in various applications. For example, in spatial transcriptomics, estimated kernel parameters are used to identify spatial variable genes, which exhibit significant expression patterns across different tissue locations. However, before these parameters can be meaningfully interpreted, it is essential to establish their identifiability. Existing studies of GP parameter identifiability have focused primarily on Matérn-type kernels, as their spectral densities allow for more established mathematical tools. In many real-world applications, particulary in time series analysis, other kernels such as the squared exponential, periodic, and rational quadratic kernels, as well as their combinations, are also widely used. These kernels share the property of being holomorphic around zero, and their parameter identifiability remains underexplored. In this paper, we bridge this gap by developing a novel theoretical framework for determining kernel parameter identifiability for kernels holomorphic near zero. Our findings enable practitioners to determine which parameters are identifiable in both existing and newly constructed kernels, supporting application-specific interpretation of the identifiable parameters, and highlighting non-identifiable parameters that require careful interpretation.

## 1 Introduction

Gaussian Processes (GPs) are powerful and flexible tools extensively used across multiple fields, such as machine learning (ML), geospatial and spatiotemporal analysis, biomedicine, finance, and environmental modeling (Rasmussen and Williams, 2006; Banerjee et al., 2014; Cressie and Wikle, 2015). They serve various purposes: as regression or classification methods through GP regression or GP classification; as priors over functions in Bayesian inference (Ghosal and van der Vaart, 2017); for modeling latent distributions via Gaussian Process Latent Variable Models (GPLVM, Lawrence (2003)); and in demonstrating equivalencies to deep neural networks with infinite width (Lee et al., 2018). The flexibility of GPs as universal approximators, their inherent interpretability – especially regarding kernel parameters – and their capability to quantify uncertainty, are among their key advantages.

The kernel function, also known as the covariance function or covariogram, which defines the covariance structure within a GP, is pivotal to application and effectiveness. Over recent decades, there has been a proliferation of research into developing specialized kernels tailored for specific data types including time-series, spatial, imaging, and spatiotemporal datasets. Popular choices such as the squared exponential (SE, also known as RBF or Gaussian), rational quadratic (RQ), periodic (Per), and Matérn kernels are frequently employed, often in innovative combinations that enhance model performance (Wang et al., 2018). These combinations involve operations such as summation, multiplication, and spectral mixtures (Duvenaud et al., 2011; 2013; Kronberger and Kommenda, 2013; Wilson and Adams, 2013; Samo and Roberts, 2015; Remes et al., 2017; Cheng et al., 2019; Verma

and Engelhardt, 2020), which enable the leveraging of individual kernel strengths to better capture complex data patterns.

Despite the extensive literature on GP theory and its application to regression or prediction tasks, less attention has been paid to the parameter inference, particularly the identifiability and interpretability of kernel parameters. Parameter inference is critical in applications that use estimated parameters in downstream tasks such as model comparison and problem-specific parameter interpretation.

One such application is in the study of spatial transcriptomics, which measures gene expression across different tissue locations to understand cellular and tissue-level biological processes (Marx, 2021). One important task within this field is identifying spatially variable genes (SVGs), which are genes that show significant changes in expression patterns across spatial locations, among tens of thousands of genes. Svensson et al. (2018) models gene expressions as a GP across spatial coordinates using a SE kernel with nuggets: $K(x, x') = \sigma_s^2 \exp\left(-\frac{\|x - x'\|^2}{2\ell^2}\right) + \sigma_e^2 1_{\{x=x'\}}$. The kernel parameter $\sigma_s^2$ was then interpreted as the magnitude of the spatial effects to identify SVGs, estimated by the Maximum Likelihood Estimator (MLE). Other applications of GP parameter inference to spatial transcriptomics include Weber et al. (2023) and Sun et al. (2020).

Another example where kernel parameter estimates are interpreted is the decomposition of the Mauna Loa $CO_2$ time series data (Tans and Keeling, 2023) into four kernel components in the impactful book Rasmussen and Williams (2006):

$$
\begin{aligned}
K(x, x') &= K_1(x, x') + K_2(x, x') + K_3(x, x') + K_4(x, x') \\
&= \theta_1^2 \exp\left(-\frac{(x - x')^2}{2\theta_2^2}\right) + \theta_3^2 \exp\left(-\frac{(x - x')^2}{2\theta_4^2} - \frac{2\sin^2(\pi(x - x'))}{\theta_5^2}\right) \\
&\quad + \theta_6^2 \left(1 + \frac{(x - x')^2}{2\theta_8 \theta_7^2}\right)^{-\theta_8} + \theta_9^2 \exp\left(-\frac{(x - x')^2}{2\theta_{10}^2}\right) + \theta_{11}^2 1_{\{x=x'\}}
\end{aligned}
\tag{1}
$$

where $K_1(x, x')$ is a SE kernel that captures the long-term smooth rising trend, $K_2(x, x')$, called the damped periodic kernel, is a multiplication of a SE kernel and a periodic kernel that accounts for seasonal variations, $K_3(x, x')$ is a rational quadratic kernel that models medium-term irregularities, and $K_4(x, x')$ is a sum of a SE and a white noise kernel that measure correlated and independent noise respectively. This kernel is also used as an example in the tutorial of the widely used Python package "sklearn.gaussian_process", with detailed interpretation of all 11 parameters proposed by the authors in Section 3. Although the interpretation seems reasonable, a theoretical understanding with a rigorous proof is missing.

Although parameter identifiability in a GP model might seem straightforward at first glance, it is a challenging and nuanced problem. In fact, not all parameters in widely used GP kernels are identifiable: if a parameter is not identifiable, consistent estimation and subsequent interpretation are impossible. For example, for the Matérn kernel in dimension $p \le 3$ with spatial variance $\sigma^2$, lengthscale $\ell$, and known smoothness parameter $\nu$, Zhang (2004) proved that neither $\sigma^2$ nor $\ell$ is identifiable or consistently estimable, no matter how sophisticated the estimator is. In fact, the only identifiable parameter in the Matérn kernel, termed the microergodic parameter, is $\sigma^2 \ell^{-2\nu}$. Follow-up studies for a single Matérn kernel include Anderes (2010); Kaufman and Shaby (2013); Li (2022); Li et al. (2023), and Chen et al. (2024) for a linear combination of Matérns with different smoothness. Such negative results raise a natural question: are all parameters used in practice, including those in Equation (1), identifiable so that their interpretations are justified?

As far as we are aware, identifiability of the parameters in Equation (1) has not been proven before. More importantly, there is still a gap in the literature for more complicated kernel combinations like those popularized in ML, especially when the combinations involve periodic kernels. The lack of theoretical examination is partly due to the failure of traditional methods used to study GP parameter identifiability, such as the integral test (Stein, 1999), which requires conditions on the spectral density not met by common kernels like SE, Per, and RQ, and even more so when these kernels are combined. This necessitates the development of new analytic tools to better understand kernel parameter identifiability and interpretability.

Motivated by these observations and challenges, this paper proves a general theorem (Theorem 3.4) that determines all the identifiable functions of the parameters in any family of stationary kernels holomorphic around 0. The result applies to complex combinations of kernels, particularly those common in the ML community, such as the one used in Equation (1). We demonstrate that all parameters in this kernel are identifiable under mild constraints, supporting the interpretation of kernel parameters in Rasmussen and Williams (2006) and the "sklearn.gaussian_process" Python package tutorial. Additionally, we establish a general result that is used to determine the identifiable functions of the parameters for a kernel that is a sum of products of other kernels.

The paper is organized as follows. Section 2 provides a comprehensive background, introduces the necessary notation and concepts, and reviews the relevant literature. Section 3 presents our main theoretical contributions. Section 4 contains simulation studies to support our theories, followed by Section 5 with a discussion of limitations and future work. A brief discussion of the connection between parameter identifiability and prediction is given in Appendix C.

## 2 BACKGROUND

This section defines key concepts and notations and summarizes existing literature on GP kernel identifiability and interpretability. We begin with the definition of GPs.

### 2.1 GAUSSIAN PROCESS

**Definition 1** (GP). *A stochastic process $Z$ is said to follow a GP in domain $\Omega$ with a mean function $\mu : \Omega \to \mathbb{R}$ and a positive definite covariance/kernel/covariogram function $K : \Omega \times \Omega \to \mathbb{R}$ if for all $x_1, \cdots, x_n \in \Omega$,*

$$[Z(x_1, ), \cdots, Z(x_n)] \sim N(v, \Sigma), \ v = [\mu(x_1), \cdots, \mu(x_n)], \ \Sigma_{ij} = K(x_i, x_j).$$

For our study, as well as presentation simplicity, we assume $\mu = 0$, without loss of generality (Stein, 1999). In this situation, since the distribution of $Z$ is completely determined by $K$, we sometimes call $K$ a GP, which refers to a GP with covariance kernel $K$.

Throughout this paper, we focus on the infill domain (also known as fixed domain or interpolation), i.e., the domain $\Omega = [0, T]^p$ does not grow with sample size, a situation commonly considered in the literature (Stein, 1999). Next, we introduce the commonly accepted stationarity assumption:

**Definition 2** (Stationarity). *$K$ is called stationary if $K(x, x') = K(x + h, x' + h), \ \forall h \in \mathbb{R}^p$.*

For a stationary kernel, we can reformulate the kernel to a function on $\mathbb{R}^p$ instead of $\mathbb{R}^p \times \mathbb{R}^p$ by $K_0(x) \coloneqq K(x, 0)$. Stationarity is a common assumption in GP literature due to its satisfactory practical performance and simplicity in both implementation and theoretical analysis. Throughout this paper, we focus only on stationary kernels, and still denote the simplified kernel $K$ without causing any confusion.

### 2.2 KERNELS

We first note that all kernel functions considered in this paper are continuous functions unless noted otherwise. Then we introduce the following commonly used kernels in Table 1.

In this table, $\sigma^2$ is called the spatial variance, or partial sill, which measures point-wise variance; $\ell$ is called the length scale that measures the spatial dependency; $\gamma > 0$ is the period parameter; $\alpha$ is called the scale mixture parameter; $\nu$ is called the smoothness parameter. Among them, Per is well-defined only when $p = 1$, i.e., $\Omega = [0, T]$ is a closed interval, while others are well-defined on $\mathbb{R}^p$ for any $p$.

Each individual kernel in the above table captures some unique behavior in the process $Z$. However, when the process has complicated structure, a common approach is to combine some of these kernels to create a new one. Such a combination can be simply a sum of products of these kernels, which is guaranteed to be a positive definite function.

Table 1: Example kernels, parameters, and domain dimension

| Name | $K(x)$ | Parameters | Dimension |
|------|--------|------------|-----------|
| SE | $\sigma^2 \exp\left(-\frac{\|x\|^2}{2\ell^2}\right)$ | $\sigma^2, \ell > 0$ | $p \geq 1$ |
| Per | $\sigma^2 \exp\left(-\frac{2\sin^2\left(\frac{\pi x}{\gamma}\right)}{\ell^2}\right)$ | $\sigma^2, \ell, \gamma > 0$ | $p = 1$ |
| RQ | $\sigma^2 \left(1 + \frac{\|x\|^2}{2\alpha\ell^2}\right)^{-\alpha}$ | $\sigma^2, \ell, \alpha > 0$ | $p \geq 1$ |
| Matérn | $\sigma^2 \frac{2^{1-\nu}}{\Gamma(\nu)} \left(\sqrt{2\nu}\frac{\|x\|}{\ell}\right)^{\nu} K_\nu\left(\sqrt{2\nu}\frac{\|x\|}{\ell}\right)$ | $\sigma^2, \ell, \nu > 0$ | $p \geq 1$ |

The following example extends Equation (1) used in Rasmussen and Williams (2006) to study the Mauna Loa $CO_2$ time series data:

$$K_\theta(x) = \theta_1^2 \exp\left(-\frac{x^2}{2\theta_2^2}\right) + \theta_3^2 \exp\left(-\frac{x^2}{2\theta_4^2} - \frac{2\sin^2\left(\frac{\pi x}{\gamma}\right)}{\theta_5^2}\right)$$

$$+ \theta_6^2 \left(1 + \frac{x^2}{2\theta_8\theta_7^2}\right)^{-\theta_8} + \theta_9^2 \exp\left(-\frac{x^2}{2\theta_{10}^2}\right) + \theta_{11}1_{\{x=0\}}, \tag{2}$$

where $\theta = (\theta_1, \cdots, \theta_{11}, \gamma)$ be the vector of all parameters in the above kernel.

Note that this kernel is more flexible than the one in Equation (1), which assumes the period $\gamma = 1$. We adopt this more challenging modification since the period is sometimes unknown in practice so practitioners have to estimate it from the data.

## 2.3 IDENTIFIABILITY

The study of identifiability of GP kernel parameters relies on the notation of equivalence of measures defined below:

**Definition 3** (Equivalence of measures). *Two measures $P_1$ and $P_2$ are said to be equivalent if they are absolutely continuous with respect to each other, denoted by $P_1 \equiv P_2$. That is, $P_1(A) = 0 \iff P_2(A) = 0$. Two measures are said to be orthogonal, denoted by $P_1 \perp P_2$ if there exists a measurable set $A$ such that $P_1(A) = 0$ but $P_2(A) = 1$.*

Two GP laws $P_1$ and $P_2$ are either equivalent, or are orthogonal (Feldman, 1958), which means they assign probability 1 to disjoint sets: $P_1(A^c) = 1$ and $P_2(A) = 1$. We define the identifiability of GP parameters as follows:

**Definition 4** (Microergodicity). *Let $(K_\theta)_{\theta\in\Theta}$ be a family of covariance kernels of a GP. Then a function $h = h(\theta)$ of $\theta$ is said to be microergodic if $K_{\theta_1} \equiv K_{\theta_2} \iff h(\theta_1) = h(\theta_2)$.*

If $h$ and $\widetilde{h}$ are both microergodic, then $h(\theta_1) = h(\theta_2) \iff \widetilde{h}(\theta_1) = \widetilde{h}(\theta_2)$, so $h$ and $\widetilde{h}$ are related by a bijection. Thus the migroergodic function is unique up to a bijective transformation, and it makes sense to speak of 'the' microergodic function $h$.

**Definition 5** (Identifiability). *Let $(K_\theta)_{\theta\in\Theta}$ be a family of covariance kernels of a GP. A function $g = g(\theta)$ of $\theta$ is said to be identifiable if $K_{\theta_1} \equiv K_{\theta_2} \implies g(\theta_1) = g(\theta_2)$, or equivalently, $g$ is a function of the microergodic function $h$. We say that the family $(K_\theta)_{\theta\in\Theta}$ is identifiable if $\theta$ is identifiable.*

Note that a consistent estimator of $g(\theta)$ can exist only when $g(\theta)$ is identifiable – when $g(\theta)$ is not identifiable, say $K_{\theta_1} \equiv K_{\theta_2}$ with $g(\theta_1) \neq g(\theta_2)$, it is not possible to find a consistent estimator of $g(\theta)$, since there is no way to distinguish between data generated from $K_{\theta_1}$ and those from $K_{\theta_2}$ almost surely (see Stein (1999); Zhang (2004) for more detailed discussion). Thus anything that can be consistently estimated is identifiable. The microergodic function $h(\theta)$ is the maximal identifiable function, so knowing the microergodic function completely solves the identifiability problem for the family of kernels. However, in some cases, it is difficult to fully determine the microergodic function $h(\theta)$, whereas it is easier to determine that some specific function $g(\theta)$ is identifiable.

To study the identifiability of GP kernel parameters, it suffices to determine when two GPs in the same parametric family are equivalent. However, to determine whether two GPs are equivalent is not an easy task, and the methods for doing so highly depend on the form of the kernels. There is a rich literature focusing on identifiability for Matérn kernels, where it has been shown that

$$K_{\sigma_1^2,\ell_1,\nu_1} \equiv K_{\sigma_2^2,\ell_2,\nu_2} \iff \begin{cases} \left(\sigma_1^2\ell_1^{-2\nu_1}, \nu_1\right) = \left(\sigma_2^2\ell_2^{-2\nu_2}, \nu_2\right) & 1 \le p \le 3, \\ \left(\sigma_1^2, \ell_1, \nu_1\right) = \left(\sigma_2, \ell_2, \nu_2\right) & p \ge 5. \end{cases}$$

That is, when the domain dimension is greater than or equal to 5, then $h(\sigma^2, \ell, \nu) = (\sigma^2, \ell, \nu)$, so all three parameters are identifiable (Anderes, 2010; Bolin and Kirchner, 2023); when the domain dimension is less than or equal to 3, then $h(\sigma^2, \ell, \nu) = \left(\sigma^2\ell^{-2\nu}, \nu\right)$: $\nu$ is identifiable (Loh et al., 2021), but not $\sigma^2$ or $\ell$ (Zhang, 2004). As a result, there is no consistent estimator of $\sigma^2$ or $\ell$, but instead, a consistent estimator of $\sigma^2\ell^{-2\nu}$, called the microergodic parameter, does exist (Kaufman and Shaby, 2013; Loh et al., 2021), namely the MLE. The microergodic function for $p = 4$ is an open problem.

Although the identifiability of Matérn has been understood, the study of other kernels including Per and RQ is much sparser. The key reason is that the tool to study equivalence between Matérn kernels, known as the integral test (Stein, 1999), requires strong conditions on the spectral densities of the kernel, which are not often satisfied by other kernels. The spectral density is defined below:

**Definition 6** (Spectral measure). *For a stationary kernel $K$, its spectral measure, denoted by $F$, is defined through*

$$K(x) = \int e^{i\omega^\top x} F(d\omega).$$

*Bochner's theorem guarantees the existence and uniqueness of $F$. The density of $F$ w.r.t. the Lebesgue measure $d\omega$, denoted by $f$, if it exists, is called the spectral density.*

The condition to use the integral test is that $f(\omega)\|\omega\|^\alpha \asymp 1$ as $\omega \to \infty$ for some $\alpha > 0$. That is, the spectral density is required to behave like $\|\omega\|^{-\alpha}$ for some positive $\alpha > 0$. The spectral density of Matérn is $f(\omega) = \sigma^2 \frac{2^p \pi^{-p/2} \Gamma\left(\nu + \frac{p}{2}\right)(2\nu)^\nu}{\Gamma(\nu)\ell^{2\nu}} \left(\frac{2\nu}{\ell^2} + \|\omega\|^2\right)^{-\left(\nu + \frac{p}{2}\right)} \asymp \|\omega\|^{-2\nu-p}$ (Rasmussen and Williams, 2006, p. 84) (note that we use a different Fourier transform convention than (Rasmussen and Williams, 2006).) However, this condition is not met by RBF, Per, or RQ, as their spectral densities decay very rapidly due to the infinite differentiability of the kernels.

Due to the popularity of these kernels in ML, we aim to address these challenges and study the equivalence of GPs with these kernels and their combinations. The next section provides theoretical support for the success of these kernels in terms of identifiability and interpretability.

## 3 THEORY

In this section, we present our main theory regarding equivalence of GPs, as outlined in the previous sections. We first determine the identifiable parameters of the individual kernels used in Equation (2), i.e., SE, Per, and RQ, with some extensions.

**Theorem 3.1.** *The microergodic functions of 5 individual kernels in Table 1, including all four components $K_1, K_2, K_3, K_4$ in Equation (2) and an additional kernel, Cosine, are summarized in Table 2.*

Theorem 3.1 supports the identifiability and interpretability of each kernel parameter in SE, Per and RQ, as discussed in Section 2.2. In addition, we include the cosine kernel, which will be revisited later in this section.

Then we consider the combination of SE, PER and RQ in Equation (2), an extension of the kernel used by the impactful book Rasmussen and Williams (2006) and the tutorial of the widely used Python package "sklearn.gaussian_process".

Table 2: Microergodic functions of five kernels

| Name | $K(x)$ | Parameters | Microergodicity | $p$ |
|---|---|---|---|---|
| SE | $\sigma^2 \exp\left(-\frac{\|x\|^2}{2\ell^2}\right)$ | $\sigma^2, \ell > 0$ | $(\sigma^2, \ell)$ | $\geq 1$ |
| Per | $\sigma^2 \exp\left(-\frac{2\sin^2\left(\frac{\pi x}{\gamma}\right)}{\ell^2}\right)$ | $\sigma^2, \ell, \gamma > 0$ | $(\sigma^2, \ell, \gamma)$ | $1$ |
| Damped Per | $\sigma^2 \exp\left(-\frac{x^2}{2\ell_1^2} - \frac{2\sin^2\left(\frac{\pi x}{\gamma}\right)}{\ell_2^2}\right)$ | $\sigma^2, \ell_1, \ell_2, \gamma > 0$ | $(\sigma^2, \ell_1, \ell_2, \gamma)$ | $1$ |
| RQ | $\sigma^2 \left(1 + \frac{\|x\|^2}{2\alpha\ell^2}\right)^{-\alpha}$ | $\sigma^2, \alpha, \ell > 0$ | $(\sigma^2, \alpha, \ell)$ | $\geq 1$ |
| Cosine | $\sigma^2 \cos(s^\top x)$ | $\sigma^2, s_1 > 0, s \in \mathbb{R}^p$ | $s$ | $\geq 1$ |

**Theorem 3.2.** *All parameters in Equation (2) are identifiable provided $\theta_{10}$, the length-scale of the SE component to model the correlated noise, is less than $\theta_2$, the length-scale of the SE component to model the long-term trend.*

Such a constraint is necessary, and not surprising, since otherwise, say, if $\theta_2 = \theta_{10}$, then we can merge the two SE components into a single SE: $(\theta_1^2 + \theta_9^2) \exp\left(-\frac{x^2}{2\theta_2^2}\right)$, making $\theta_1^2 + \theta_9^2$ identifiable instead of $\theta_1^2$ and $\theta_9^2$. This distinction of two SE components is also discussed in Section 5.4.3 in Rasmussen and Williams (2006). Excluding this trivial case, all parameters are identifiable. As a consequence, these parameters are interpretable, as discussed in the same section in Rasmussen and Williams (2006). For example, $\theta_1$ measures the amplitude and $\theta_2$ measures the characteristic length-scale of the long-term smooth rising trend; within the seasonal trend, $\theta_3$ gives the magnitude, $\theta_4$ gives the decay time for the periodic component, $\gamma$ gives the period, while $\theta_5$ is the smoothness of the periodic component; for the (small) medium term irregularities, $\theta_6$ is the magnitude, $\theta_7$ is the typical length-scale and $\theta_8$ is the shape parameter determining diffuseness of the length-scales; $\theta_9$ is the magnitude of the correlated noise component, $\theta_{10}$ is its lengthscale and $\theta_{11}$ is the magnitude of the independent noise component.

Now we would like to answer the following more challenging question with a broader implication: Given a new kernel, how do we determine the microergodic function? Specifically, if we combine a finite number of kernels, such as those in Table 2, by finite multiplication and addition like Equation (2), what is the microergodic function of the resulting kernel? To answer these questions, we need to introduce the following notions first.

**Lemma 3.3** (Kernel decomposition). *For any stationary kernel $K$, $K$ can be uniquely decomposed as $K = K^c + K^d$, where $K^c$ is a kernel with continuous spectral measure and $K^d$ is another kernel with discrete spectral measure.*

A direct consequence is that if $K$ admits a spectral density, then $K = K^c$; while if $K$ is periodic, then $K = K^d$. We call $K^c$ the continuous component and $K^d$ the discrete component. Note that this notion is different from continuous and discrete functions, and we do assume all kernels are continuous functions themselves. Here the continuous and discrete notion is at the spectrum level. For example, the Per kernel, is continuous as a function, but has a purely discrete spectrum. Moreover, we denote the spectral measure of $K^c$ as $F^c$ and the spectral measure of $K^d$ as $F^d$, where $F = F^c + F^d$ is the spectral measure of $K$.

Such a decomposition offers deeper insights to understand different types of kernels. Moreover, to understand the equivalence of GPs, it suffices to understand the equivalence of its continuous component and discrete component separately, given by the following key theorem, which is the main result of the paper:

**Theorem 3.4.** *Given two kernels $K_1$ and $K_2$ with $K_1$ holomorphic on some ball around $0$ in $\mathbb{C}^p$, the $p$-dimensional complex space, then $K_1 \equiv K_2$ if and only if the following two conditions hold:*

*1. $K_1^c(x) = K_2^c(x)$ for every $x \in \mathbb{R}^p$.*

2. *There are $c, C > 0$ such that $cF_1^d(\{\omega\}) \le F_2^d(\{\omega\}) \le CF_1^d(\{\omega\})$ for all $\omega \in \mathbb{R}^p$ and*
$$\sum_{\omega: F_1^d(\{\omega\}) > 0} \left(1 - \frac{F_2^d(\{\omega\})}{F_1^d(\{\omega\})}\right)^2 < \infty.$$

Note that $K : \mathbb{R}^p \to \mathbb{R}$ is said to be holomorphic on a ball around $0$ in $\mathbb{C}^p$, if it has a holomorphic extension $\tilde{K}$ to some ball $B \subset \mathbb{C}^p$ around $0$, such that $\tilde{K} = K$ on $B \cap \mathbb{R}^p$. While being holomorphic on a ball around $0$ is a stronger condition than being infinitely differentiable, most infinitely differentiable kernels used in practice, including all those in Table 2, are holomorphic on a ball around $0$. Condition 1 means the continuous components of $F_1$ and $F_2$ are the same, while Condition 2 means the discrete components of $F_1$ and $F_2$ have the same support, and their relative difference, although allowed to be nonzero, should decay fast enough.

Notably, Theorem 3.4 provides a general pipeline to study the identifiability of kernel parameters, summarized in the following theorem:

**Theorem 3.5.** *Let $(K_\theta)_{\theta \in \Theta}$ be a family of stationary kernels on $\mathbb{R}^p$, each of which is holomorphic on some ball around $0$ in $\mathbb{C}^p$. We have the following assertions regarding the microergodic function:*

1. *If $h(\theta)$ is microergodic for the continuous component $(K_\theta^c)_{\theta \in \Theta}$ and $g(\theta)$ is microergodic for the discrete component $(K_\theta^d)_{\theta \in \Theta}$, then $(h(\theta), g(\theta))$ is microergodic for $(K_\theta)_{\theta \in \Theta}$.*

2. *Moreover,*

   (a) *$h(\theta)$ is microergodic for $(K_\theta^c)_{\theta \in \Theta}$ if and only if*
   $$K_{\theta_1}^c(x) = K_{\theta_2}^c(x), \ \forall x \in \mathbb{R}^p \iff h(\theta_1) = h(\theta_2).$$

   (b) *$g(\theta)$ is microergodic for $(K_\theta^d)_{\theta \in \Theta}$ if and only if*
   $$\left.\begin{array}{c} \exists c, C > 0, \ s.t. \ cF_1^d(\{\omega\}) \le F_2^d(\{\omega\}) \le CF_1^d(\{\omega\}), \ \forall \omega \in \mathbb{R}^p \\ \sum_{\omega: F_{\theta_1}^d(\{\omega\}) > 0} \left(1 - \frac{F_{\theta_2}^d(\{\omega\})}{F_{\theta_1}^d(\{\omega\})}\right)^2 < \infty \end{array}\right\} \iff g(\theta_1) = g(\theta_2)$$

That is, in order to find the microergodic function of a parametric family of kernels $K_\theta$, it suffices to find the microergodic function $h(\theta)$ of the continuous component, and $g(\theta)$ of the discrete component separately. Moreover, to find $h(\theta)$, it suffices to understand when two continuous components are equal everywhere; to find $g(\theta)$, we need to investigate the conditions 2b about the discrete measure $F^d$.

Having established the foundational aspects of kernel identifiability, we now apply our results to determine the microergodic function of various combinations of kernels. Our general strategy is to use Fourier transform identities to compute the spectral measure of the combined kernel (see, for example, Theorem B.6) and then apply Theorem 3.4. These combinations not only illustrate the practical applications of our theoretical findings in Theorem 3.4, but also provide insights into designing new kernels with desired properties. We start with the squared exponential kernel with automatic relevance determination (ARD).

**Theorem 3.6.** *For the family*
$$K_{\sigma, M}(x) = \sigma^2 \exp\left(-\frac{1}{2}x^T M x\right),$$

*where $\sigma^2 > 0$ and $M$ is a positive-definite matrix, the microergodic function is $(\sigma^2, M)$.*

Next, we study the sum of cosine kernels:

**Theorem 3.7.** *For the family*
$$K_{\sigma_1, \dots, \sigma_m, s_1, \dots, s_m}(x) = \sigma_1^2 \cos(s_1 x) + \sigma_2^2 \cos(s_2 x) + \cdots + \sigma_m^2 \cos(s_m x),$$

*where $\sigma_1^2, \dots, \sigma_m^2 > 0$ and $s_1, \dots, s_m \in \mathbb{R}$, under the natural constraint $0 \le s_1 < s_2 < \cdots < s_m$, the microergodic function is $(s_1, \dots, s_m)$.*

Theorem 3.7 shows that when cosine kernels are combined linearly, their individual frequencies (or periods) remain identifiable, provided they are distinct. This scenario often arises in signal processing where different periodic components need to be isolated and identified. Notably, the last kernel in Table 1 is a special case of the kernel in Theorem 3.7 with $m = 1$.

Next, we study the product of Cosine kernels:

**Theorem 3.8.** *For the family*

$$K_{\sigma,s_1,\ldots,s_m}(x) = \sigma^2 \cos(s_1 x)\cos(s_2 x)\cdots\cos(s_m x),$$

*where $\sigma^2 > 0$ and $s_1,\ldots,s_m \in \mathbb{R}$, under the natural constraint $0 < s_1 \leq s_2 \leq \cdots \leq s_m$, the microergodic function is $\{\pm s_1 \pm s_2 \pm \cdots \pm s_m\} := \{a_1 s_1 + a_2 s_2 + \cdots + a_m s_m : a_1, a_2, \ldots, a_m \in \{-1,1\}\}$. If $m = 1, 2, 3$, then the mircoergodic function simplifies to $(s_1,\ldots,s_m)$.*

In Theorem 3.8, when $m \geq 4$, we do not have identifiability of $(s_1,\ldots,s_m)$. For example, for $m = 4$, when $(s_1, s_2, s_3, s_4) = (1, 2, 2, 3)$ and $(\widetilde{s}_1, \widetilde{s}_2, \widetilde{s}_3, \widetilde{s}_4) = (1, 1, 3, 3)$, the values of the microergodic function coincide, that value being $\{0, \pm 2, \pm 4, \pm 6, \pm 8\}$. Theorem 3.8 shows that for a product of discrete spectrum kernels that are all a function of the same variable $x$, the parameters of each individual kernel may not be identifiable.

Finally, we explore the sum of periodic kernels as previously discussed.

**Theorem 3.9.** *Let $K_{\ell,\gamma}$ denote the periodic kernel with variance parameter 1, length-scale $\ell$, and period $\gamma$. For the family*

$$K_{\sigma_1,\sigma_2,\gamma_1,\gamma_2}(x) = \sigma_1^2 K_{\ell_1,\gamma_1}(x) + \sigma_2^2 K_{\ell_2,\gamma_2}(x),$$

*where $\sigma_1^2, \sigma_2^2, \ell_1, \ell_2 > 0$, $\gamma_1 > \gamma_2 > 0$, the microergodic function is $(\sigma_1^2, \ell_1, \gamma_1, \sigma_2^2, \ell_2, \gamma_2)$, that is, all parameters are identifiable.*

This result is crucial for scenarios where multiple periodic processes operate at different scales or periods, as often encountered in geospatial, financial, and environmental data analysis.

## 4 SIMULATION

In this section, we provide empirical support to our theoretical results on kernel parameter identifiability, presented in Section 3, by investigating the behavior of the maximum likelihood estimators (MLEs) as the sample size $n$ increases.

Before moving to the simulation details, we would like to clarify the broader picture of parameter inference for GPs, which involves three steps: first, determining which parameters are identifiable; second, finding a consistent estimator of identifiable parameters, such as the MLEs or others estimators; and third, developing numerical methods to compute these estimators. While the second and the third steps are crucial, they fall beyond the scope of this paper, which focuses solely on the first step–a theoretical framework to find all the identifiable parameters. In fact, even for simple kernels like the SE and Matérn kernels, whether the MLE is consistent remains open (Loh and Sun, 2023).

Despite these complexities, we use standard optimization packages commonly applied in the GP literature to find the MLEs. Our simulations are not intended to solve the open problem of MLE consistency or introduce new numerical techniques; rather, they serve to illustrate the theoretical results on identifiability through practical examples.

We start from individual kernels, followed by the combination in Equation (2).

### 4.1 INDIVIDUAL KERNELS

We consider the individual kernels: SE, Damped Per (DPer), Per, RQ, and Cosine. For the cosine kernel, we parameterize in terms of the period $\gamma$ so that $s = \frac{2\pi}{\gamma}$. Input samples are generated by adding a unif$(-\frac{1}{4n}, \frac{1}{4n})$ random shift to $n$ evenly spaced points in $[\frac{1}{4n}, 1 - \frac{1}{4n}]$, where $n \in \{500, 1000, 2000, 5000\}$. After generating the outcomes by sampling a GP with the given kernel at the inputs, we added independent Gaussian noise from $N(0, \varepsilon)$, $\varepsilon = 0.01$, to model measurement errors (see Section D of the appendix for the experiments repeated with

$\varepsilon = 0.1$). All kernel parameters were estimated by MLEs, with 100 replicates for each kernel configuration to assess the convergence of the MLEs. The results are summarized in Figure 1. These boxplots demonstrate that the MLEs of all parameters except $\sigma^2$ in the cosine kernel appear consistent, as conjectured by their identifiability, proved in Theorem 3.1.

Some of the MLE standard deviations appear to plateau for large $n$. One explanation for this is numerical limitations – for our squared exponential simulation, where $\sigma^2 = 1$ and $\ell = 1/500$, the condition numbers of the covariance matrix of the observations are $6.82 \cdot 10^1$, $1.80 \cdot 10^8$, $1.89 \cdot 10^{19}$, and $3.37 \cdot 10^{24}$ for sample sizes $500, 1000, 2000,$ and $5000$, respectively.

The failure of the MLE of $\sigma^2$ in the cosine kernel to converge is in agreement with the microergodicity of $\gamma$. In fact, if we treat $\gamma$ as known and let the noise variance $\varepsilon$ decrease to 0, then since the covariance matrix has rank 2 for all $n \geq 2$, it can be shown that the MLE $\widehat{\sigma}(\varepsilon)$ converges to $\widehat{\sigma}(0) \sim \sigma^2 \frac{\chi_2^2}{2}$.

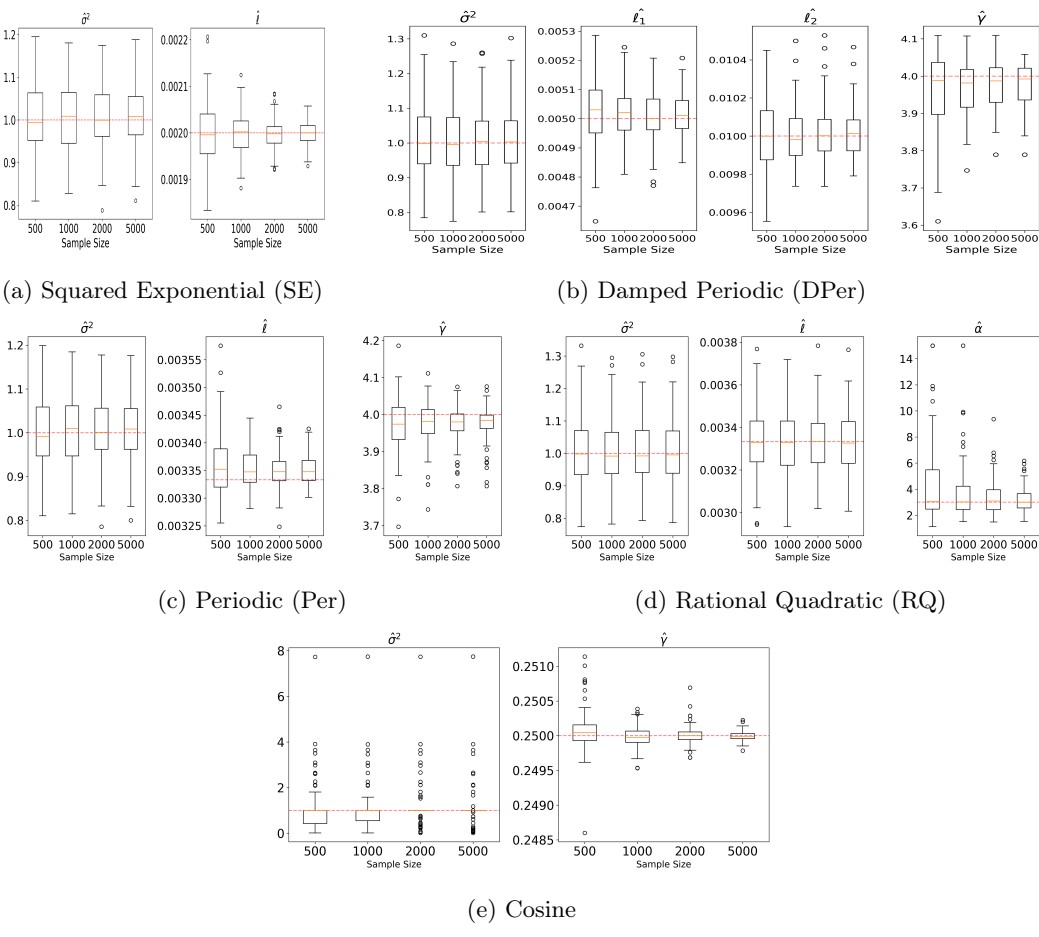

Figure 1: Simulation results for various kernel types. Each subfigure shows the boxplots of MLEs for the corresponding kernel, with ground truth in horizontal dashed line.

## 4.2 The combined kernel in Equation (2)

Then, we study the combined kernel, one motivating kernel of this paper, defined in Equation (2). Since the kernel was proposed for forecasting $CO_2$level on the Mona Loa dataset, we set the time interval to be $[0, 45]$, presenting the time span of 45 years. Input samples are generated by adding a unif$(-\frac{45}{4n}, \frac{45}{4n})$ random shift to $n$ evenly spaced points in $[\frac{45}{4n}, 45 - \frac{45}{4n}]$, where $n \in \{50, 100, 200, 500\}$. All kernel parameters were estimated by MLEs, with 100 replicates to assess the convergence of the MLEs. Moreover, to further mimic this dataset,

the ground truth parameters and noise variance $\theta_{11}^2$ are set to be the MLEs learned from running the "Gaussian process regression" package from the scikit-learn Python package. All truth parameters to be estimated are given by Table 3.

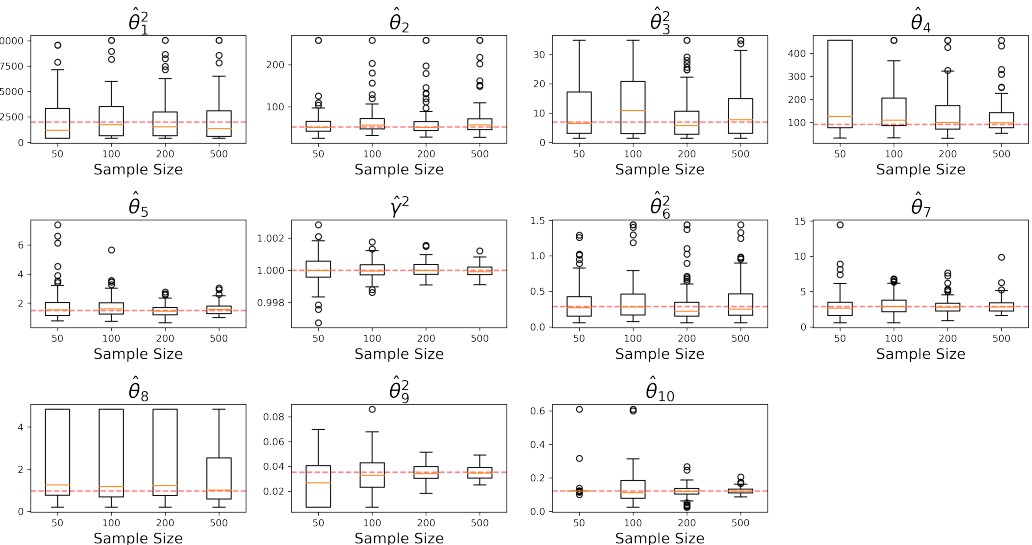

Figure 2: MLEs of parameters in Equation (2), with ground truth in horizontal dashed line.

In Figure 2, we again observe that the MLEs generally are unbiased, but for some parameters, their variance does not strictly decrease with sample size. This is likely due to the relatively large number of parameters (10) compared to the small sample size of 500.

## 5 Discussion

This paper has introduced a novel analytical framework that advances the theory of identifiability of kernel parameters in GPs for a large class kernels, those holomorphic around 0. We have demonstrated that all the parameters in certain combinations of kernels, such as the example employed on the Mauna Loa $CO_2$ time series data, are indeed identifiable. This establishes a robust theoretical foundation for selecting or constructing GP kernels and determining the identifiable functions of the parameters in practical applications.

Looking ahead, several avenues of future research present themselves as particularly promising and interesting. First, while establishing the identifiability of kernel parameters is a critical step, it does not necessarily guarantee the consistency of the MLE. The analysis of MLEs is complicated due to the complex nature of the likelihood function involved, which is often multi-modal and difficult to handle. Second, extending our theoretical framework to encompass non-stationary kernels could enhance the flexibility of GPs in modeling data with evolving trends and dynamics. This area is notably challenging due to the current limitations in mathematical tools available, presenting a largely open problem in the field. Third, another intriguing direction for research involves extending our findings to infinitely differentiable kernels that are not holomorphic near 0, though most infinitely differentiable kernels used in applications are holomorphic near 0.

**Reproducibility Statement**: All code used to produce the results of this paper are provided Appendix A. Complete proofs of all lemmas and theorems stated in the paper are provided in Appendix B.

**Ethics Statement**: Our paper does not deal with sensitive experiments, data, or any methods that can be expected to cause harm. We have no conflicts of interest and have no data privacy concerns.

**Acknowledgment**: AQ was supported by NIH grant R37 AI029168; DL was supported by NIH grants P30 ES010126, R01 HL149683, R01 HL173044, R01 LM014407, R56 LM013784, UM1 TR004406.

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

## APPENDICES

## A CODE AVAILABILITY

All codes can be found and downloaded at https://github.com/Ameer-eng/iclr2025-simulation.

## B PROOFS

We begin with the proof of Lemma 3.3, Theorem 3.4, and Theorem 3.5, which are the general tools we developed to study equivalence of GPs on which the other theorems depend.

### B.1 PROOFS OF LEMMA 3.3, THEOREM 3.4, AND THEOREM 3.5

We first define the measure associated to a GP $Z$.

**Definition 7.** *For a set $\Omega$, covariance kernel $K : \Omega \times \Omega \to \mathbb{R}$ and mean function $m : \Omega \to \mathbb{R}$, define the measure $G_\Omega(m, K)$ to be the distribution of a Gaussian process $(Z(x))_{x \in \Omega}$ with mean $m$ and covariance $K$. Thus $G_\Omega(m, K)$ is a measure on $(\mathbb{R}^\Omega, \mathcal{A}_\Omega)$, where $\mathcal{A}_\Omega$ is the $\sigma$-algebra generated by the coordinate maps $Z(x)$ for $x \in \Omega$.*

Now we state Bochner's Theorem, which is the key tool for working with stationary kernels.

**Lemma B.1** (Bochner's Theorem (Stein, 1999))**.** *A continuous complex-valued function $K : \mathbb{R}^p \to \mathbb{C}$ is the covariance function of a complex valued stationary process on $\mathbb{R}^p$ if and only if there is a positive finite measure $F$ on $\mathbb{R}^p$ such that $K$ can be expressed as*

$$K(x) = \int e^{i\omega^\top x} F(d\omega).$$

*$K$ is real valued if and only if the distributions of $\omega$ and $-\omega$ are the same under $F$.*

**Now we prove Lemma 3.3**.

*Proof of Lemma 3.3.* Let $F$ be the spectral measure of $K$. Let $\Omega = \{\omega \in \mathbb{R}^p : F(\{\omega\}) > 0\}$. We claim that $\Omega$ is at most countable. For contradiction, suppose $\Omega$ is uncountable. Then $\{\omega : F(\{\omega\}) > 1/n\}$ must be uncountable for some $n$. But then $F(\mathbb{R}^p) \geq \frac{1}{n}|\{\omega : F(\{\omega\}) > 1/n\}| = \infty$, contradicting finiteness of $F$. Let

$$F^d = \sum_{\omega \in \Omega} F(\{\omega\})\delta_\omega,$$

$$F^c = F - F^d.$$

Then let $K^d$ be the inverse Fourier transform of $F^d$, i.e. $K^d(x) = \int e^{i\omega^T x} F^d(d\omega)$, and let $K^c$ be the inverse Fourier transform of $F^c$. By Bochner's theorem, both $K^c$ and $K^d$ are complex-valued stationary covariance kernels. Since $F$ is symmetric, so is $F^d$. Hence $F^c = F - F^d$ is also symmetric. Thus $K^c$ and $K^d$ are real valued. $\qquad\square$

Now we prove a condition under which we can recover a process $(Z(t))_{t \in \mathbb{R}^p}$ from it's values on a box $[0, T]^p$.

**Lemma B.2.** *Suppose $K$ is a stationary covariance kernel on $\mathbb{R}^p$ that is holomorphic on some ball containing $0$. For $\Omega \subset \mathbb{R}^p$, let*

$$\mathcal{A}_\Omega = \sigma(Z(t) : t \in \Omega).$$

*Then for any $T > 0$,*

$$\mathcal{A}_{\mathbb{R}^p} \subset \mathcal{A}^*_{[0,T]^p},$$

*where $\mathcal{A}^*_{[0,T]^p}$ denotes the completion of $\mathcal{A}_{[0,T]^p}$ by sets of probability $0$ with respect to $P = G_{\mathbb{R}^p}(0, K)$. In other words, the entire process $(Z(t))_{t \in \mathbb{R}^p}$ can be recovered with probability $1$ from $(Z(t))_{t \in [0,T]^p}$.*

*Proof.* For ease of notation, assume $p = 1$. The proof for $p > 1$ is the same. Say $K$ is holomorphic on $D(0, 2\delta)$, $\delta > 0$. By Theorem 2 of Lukacs et al. (1952), $\int e^{t\omega} F(d\omega) < \infty$ for $|t| < 2\delta$. For $k \in \mathbb{N}$, since $\frac{\delta^{2k} \omega^{2k}}{(2k)!} \leq e^{\delta\omega} + e^{-\delta\omega}$,

$$\int \omega^{2k} F(d\omega) \leq \frac{(2k)!}{\delta^{2k}} \int (e^{\delta\omega} + e^{-\delta\omega}) F(d\omega) < \infty.$$

Thus the process $Z$ is infinitely mean-square differentiable.

For $s, t \in \mathbb{R}$,

$$(Z(s), Z(t))_{L^2(P)} = \text{Cov}(Z(s), Z(t)) = K(s - t)$$
$$= \int e^{i\omega(s-t)} F(d\omega) = (e^{i\omega s}, e^{i\omega t})_{L^2(F)}.$$

Hence the map $Z(t) \mapsto e^{i\omega t}$ extends by linearity and continuity to a unitary isomorphism $\Phi$ from $H(\mathbb{R})$ to $L_{\mathbb{R}}(F)$, where $H(\mathbb{R})$ is the closure of span$\{Z(t) : t \in \mathbb{R}\}$ in $L^2(P)$, and $L_{\mathbb{R}}(F)$ is the closure of span$\{e^{i\omega t} : t \in \mathbb{R}\}$ in $L^2(F)$.

Let $t \in \left[-\frac{\delta}{2}, 0\right]$. We claim that

$$Z(t) = \sum_{n=0}^{\infty} \frac{Z^{(n)}(0)}{n!} t^n,$$

where the sum converges in $L^2(P)$. Using the unitary isomorphism above and differentiating with respect to $t$ yields

$$\Phi Z(t) = e^{i\omega t},$$
$$\Phi Z'(t) = i\omega e^{i\omega t},$$
$$\vdots$$
$$\Phi Z^{(n)}(t) = (i\omega)^n e^{i\omega t}.$$

Thus to show that $\sum_{n=0}^{\infty} \frac{Z^{(n)}(0)}{n!} t^n = Z(t)$ in $L^2(P)$, it is equivalent to show that $\sum_{n=0}^{\infty} \frac{(i\omega)^n}{n!} t^n = e^{i\omega t}$ in $L^2(F)$. We know $\sum_{n=0}^{N} \frac{(i\omega)^n}{n!} t^n \to e^{i\omega t}$ pointwise as $N \to \infty$, and

$$\left| \sum_{n=0}^{N} \frac{(i\omega)^n}{n!} t^n \right| \leq \sum_{n=0}^{N} \frac{|\omega|^n}{n!} |t|^n$$
$$\leq e^{|\omega||t|} \leq e^{\frac{\delta}{2}|\omega|} \leq e^{-\frac{\delta}{2}\omega} + e^{\frac{\delta}{2}\omega}.$$

Since, as mentioned earlier, $e^{-\frac{\delta}{2}\omega}, e^{\frac{\delta}{2}\omega} \in L^2(F)$, the dominated convergence theorem yields $\sum_{n=0}^{N} \frac{(i\omega)^n}{n!} t^n \to e^{i\omega t}$ in $L^2(F)$ as $N \to \infty$.

Thus $\mathcal{A}_{\left[-\frac{\delta}{2}, T\right]} \subset \mathcal{A}_{[0,T]}^*$. Repeating the same argument with the process $Y(t) = Z\left(t - \frac{\delta}{2}\right)$, which has the same covariance as $Z$, we obtain $\mathcal{A}_{[-\delta, T]} \subset \mathcal{A}_{[0,T]}^*$, and by induction, $\mathcal{A}_{\left[-k\frac{\delta}{2}, T\right]} \subset \mathcal{A}_{[0,T]}^*$ for all $k \in \mathbb{N}$. Similarly, $\mathcal{A}_{\left[0, T+k\frac{\delta}{2}\right]} \subset \mathcal{A}_{[0,T]}^*$ for all $k \in \mathbb{N}$. Hence $\mathcal{A}_{\mathbb{R}} \subset \mathcal{A}_{[0,T]}^*$. $\square$

We immediately conclude the following, which reduces the problem of determining equivalence of GPs on a bounded interval to determining their equivalence on all of $\mathbb{R}^p$, which is a much simpler task.

**Lemma B.3.** *Suppose $K_0$ and $K_1$ are continuous stationary covariance kernels on $\mathbb{R}^p$ and that $K_0$ is holomorphic on some ball containing $0$. Let $T > 0$. Then*

$$G_{[0,T]^p}(0, K_0) \equiv G_{[0,T]^p}(0, K_1) \iff G_{\mathbb{R}^p}(0, K_0) \equiv G_{\mathbb{R}^p}(0, K_1).$$

*Proof.* Let $P_j = G_{\mathbb{R}^p}(0, K_j)$, $j = 0, 1$. Note $G_{[0,T]^p}(0, K_j) = P_j|_{\mathcal{A}_{[0,T]^p}}$. Suppose $G_{[0,T]^p}(0, K_0) \equiv G_{[0,T]^p}(0, K_1)$, that is, $P_0$ and $P_1$ are equivalent on $\mathcal{A}_{[0,T]^p}$. Then $P_0$ and $P_1$ are equivalent on $\mathcal{A}^*_{[0,T]^p}$, the completion of $\mathcal{A}_{[0,T]^p}$ by sets of probability 0 with respect to $P_0$ (or equivalently, with respect to $P_1$, since $P_0 \equiv P_1$ on $\mathcal{A}_{[0,T]^p}$). By Lemma B.2, $\mathcal{A}_{\mathbb{R}^p} \subset \mathcal{A}^*_{[0,T]^p}$. Consequently, $P_0$ and $P_1$ are equivalent on $\mathcal{A}_{\mathbb{R}^p}$. $\qquad\square$

Parts of the following lemma are proved in Ibragimov and Rozanov (1978), but for our application, we need a precise statement of the result. In what follows, for real valued functions $a$ and $b$ defined on a set $S$, we use $a(s) \asymp b(s)$ for $s \in S$ to mean that there exist $c, C > 0$ such that $ca(s) \leq b(s) \leq Ca(s)$ for all $s \in S$.

**Lemma B.4.** *Suppose continuous stationary kernels $K_0$ and $K_1$ on $\mathbb{R}^p$ have spectral measures $F_0$ and $F_1$. Write*

$$F_j = F_j^c + F_j^d, \quad j = 0, 1,$$

*where $F_j^d = \sum_{\omega: F_j(\{\omega\}) > 0} F_j(\{\omega\}) \delta_\omega$ is the discrete part of $F_j$, and $F_j^c = F_j - F_j^d$ is the continuous part of $F_j$. Let $P_j = G_{\mathbb{R}^p}(0, K_j)$, $j = 0, 1$. For $j = 0, 1$, let $H_j(\mathbb{R}^p)$ be the closure of $\operatorname{span}\{Z(t) : t \in \mathbb{R}^p\}$ in $L^2(P_j)$. The following are equivalent:*

*(i)*

$$P_0 \equiv P_1.$$

*(ii)*

$$\| \cdot \|_{L^2(P_0)} \asymp \| \cdot \|_{L^2(P_1)} \text{ on } \operatorname{span}\{Z(t) : t \in \mathbb{R}^p\},$$

$$\|I - K_1\|^2_{\operatorname{HS}(H_0(\mathbb{R}^p))} < \infty.$$

*(iii)*

$$F_0^c = F_1^c,$$

$$F_0^d(\{\omega\}) \asymp F_1^d(\{\omega\}),$$

$$\sum_{\omega: F_0^d(\{\omega\}) > 0} \left(1 - \frac{F_1^d(\{\omega\})}{F_0^d(\{\omega\})}\right)^2 < \infty.$$

*Proof.* To clarify (ii), $\operatorname{HS}(H_0(\mathbb{R}^p))$ is the space of Hilbert-Schmidt operators on $H_0(\mathbb{R}^p)$. The continuous bilinear map $K_1 : H_0(\mathbb{R}^p) \times H_0(\mathbb{R}^p) \to \mathbb{C}$ is identified with the bounded linear map $K_1 : H_0(\mathbb{R}^p) \to H_0(\mathbb{R}^p)$ by

$$(K_1 U, V)_{H_0(\mathbb{R}^p)} = K_1(U, V), \quad U, V \in H_0(\mathbb{R}^p).$$

Under this identification, $K_0 = I$.

By Theorem 7 on page 129 of Stein (1999), (i) and (ii) are equivalent.

Suppose (ii) holds. Then $H_0(\mathbb{R}^p) = H_1(\mathbb{R}^p)$ and $\| \cdot \|_{L^2(P_0)} \asymp \| \cdot \|_{L^2(P_1)}$ on $H_0(\mathbb{R}^p)$. Hence by the unitary isomorphism $\Phi : H_j(\mathbb{R}^p) \to L_{\mathbb{R}^p}(F_j)$, $\Phi Z(t) = e^{i\omega^T t}$ defined in the proof of Lemma B.2, $L_{\mathbb{R}^p}(F_0) = L_{\mathbb{R}^p}(F_1)$ and $\| \cdot \|_{L^2(F_0)} \asymp \| \cdot \|_{L^2(F_1)}$ on $L_{\mathbb{R}^p}(F_0)$.

We claim that $L_{\mathbb{R}^p}(F_0) = L^2(F_0)$. Let $f \in C_c^\infty(\mathbb{R}^p)$. Let $R$ be a rectangle containing the support of $f$. By the Fourier inversion theorem, for every $x \in \mathbb{R}^p$, $f(x) = \int_R \hat{f}(\xi) e^{i\xi^T x} \, d\xi$. The integral is a pointwise limit of Riemann sums. Each Riemann sum has absolute value bounded by $\|f\|_{L^\infty} \operatorname{Vol}(R) \in L^2(F_0)$, so by dominated convergence, the Riemann sums converge to $f$ in $L^2(F_0)$. This shows that $f \in L_{\mathbb{R}^p}(F_0)$. Since $C_c^\infty(\mathbb{R}^p)$ is dense in $L^2(F_0)$, we obtain $L_{\mathbb{R}^p}(F_0) = L^2(F_0)$.

Then the equivalence $\| \cdot \|_{L^2(F_0)} \asymp \| \cdot \|_{L^2(F_1)}$ on $L_{\mathbb{R}^p}(F_0) = L^2(F_0)$ implies $\int \phi dF_0 \asymp \int \phi dF_1$ for all $\phi \geq 0$. Thus $F_0 \equiv F_1$, so there is a density $f(\omega) = \frac{F_1(d\omega)}{F_0(d\omega)}$, and $\int \phi dF_0 \asymp \int \phi dF_1$ for

all $\phi \geq 0$ implies $f(\omega) \asymp 1$ for $F_0$ a.e. $\omega \in \mathbb{R}^p$. If $U, V \in H_0(\mathbb{R}^p)$ and $u = \Phi U$, $v = \Phi V$, then

$$
\begin{aligned}
(K_1 U, V) = K_1(U, V) \\
= \int u(\omega)\overline{v(\omega)} F_1(d\omega) \\
= \int f(\omega) u(\omega)\overline{v(\omega)} F_0(d\omega) \\
= (fu, v)_{L^2(P_0)}.
\end{aligned}
$$

Hence via the isomorphism $\Phi$, $K_1$ is identified with $M_f$, the operator of multiplication by $f$. Thus

$$
\|I - M_f\|^2_{\mathrm{HS}(L^2(F_0))} = \|I - K_1\|^2_{\mathrm{HS}(H_0(\mathbb{R}^p))} < \infty.
$$

Let

$$
\Omega = \{\omega \in \mathbb{R}^p : F_0(\{\omega\}) > 0\}.
$$

Since $F_0 = F_0^c + F_0^d$ and $F_0^c \perp F_0^d$, we have the orthogonal decomposition $L^2(F_0) = L^2(\mathbb{R}^p \setminus \Omega, F_0^c) \oplus L^2(\Omega, F_0^d)$. Thus

$$
\|I - M_f\|^2_{\mathrm{HS}(L^2(F_0))} = \|I - M_f\|^2_{\mathrm{HS}(L^2(\mathbb{R}^p \setminus \Omega, F_0^c))} + \|I - M_f\|^2_{\mathrm{HS}(L^2(\Omega, F_0^d))}.
$$

Note $I - M_f = M_{1-f}$. We claim that $1 - f(\omega) = 0$ for $F_0^c$-a.e. $\omega \in \mathbb{R}^p \setminus \Omega$. For contradiction, suppose $F_0^c(\{1-f \neq 0\} \cap \mathbb{R}^p \setminus \Omega) > 0$. Then either $F_0^c(\{1-f > 0\} \cap \mathbb{R}^p \setminus \Omega) > 0$ or $F_0^c(\{1-f < 0\} \cap \mathbb{R}^p \setminus \Omega) > 0$, or both. Say $F_0^c(\{1-f > 0\} \cap \mathbb{R}^p \setminus \Omega) > 0$. Then there is $\varepsilon > 0$ such that $F_0^c(\{1-f > \varepsilon\} \cap \mathbb{R}^p \setminus \Omega) > 0$. Let $A = \{1-f > \varepsilon\} \cap \mathbb{R}^p \setminus \Omega$. Define $m(t) = F_0^c(A \cap (-\infty, t)^p)$. Then since $m(t+h) - m(t) = F_0^c(A \cap [t, t+h)^p) \leq F_0^c([t, t+h)^p) \to F_0^c(\{t\}^p) = 0$ as $h \to 0$, $m$ is continuous. Since $m(-\infty) = 0$ and $m(\infty) = F_0^c(A)$, there is $t$ with $m(t) = \frac{1}{2} F_0^c(A)$. Thus $A$ can be divided into two sets of positive $F_0^c$ measure. Inductively, $A$ can be divided into $k$ sets of positive $F_0^c$ measure for every $k$. Thus $\dim(L^2(A, F_0^c)) \geq k$ for all $k$, so $\dim(L^2(A, F_0^c)) = \infty$. Since $((1-f)\varphi, \varphi)^2_{L^2(A, F_0^c)} = \left( \int_A (1-f)|\varphi|^2 F_0^c(d\omega) \right)^2 \geq \varepsilon \|\varphi\|^4_{L^2(F_0^c)}$ for all $\varphi \in L^2(A, F_0^c)$, we obtain

$$
\begin{aligned}
\|I - M_f\|^2_{\mathrm{HS}(L^2(\mathbb{R}^p \setminus \Omega, F_0^c))} \geq \|I - M_f\|^2_{\mathrm{HS}(L^2(A, F_0^c))} \\
\geq \varepsilon \dim(L^2(A, F_0^c)) \\
= \infty.
\end{aligned}
$$

This is a contradiction. Thus $1 - f(\omega) = 0$ for $F_0^c$ a.e. $\omega \in \mathbb{R}^p \setminus \Omega$. Since $F_1^c(d\omega) = f(\omega) F_0^c(d\omega)$, we obtain $F_1^c = F_0^c$. Thus $\|I - M_f\|^2_{\mathrm{HS}(L^2(\mathbb{R}^p \setminus \Omega, F_0^c))} = 0$. To compute $\|I - M_f\|^2_{\mathrm{HS}(L^2(\Omega, F_0^d))}$, we note that $\{F_0^d(\{\omega\})^{-1/2} 1_{\{\omega\}} : \omega \in \Omega\}$ is an orthonormal basis of $L^2(\Omega, F_0^d)$, so

$$
\begin{aligned}
\|I - M_f\|^2_{\mathrm{HS}(L^2(\Omega, F_0^d))} = \sum_{\omega \in \Omega} ((1-f) F_0^d(\{\omega\})^{-1/2} 1_{\{\omega\}}, F_0^d(\{\omega\})^{-1/2} 1_{\{\omega\}})^2_{L^2(F_0^d)} \\
= \sum_{\omega \in \Omega} \left( \int_{\{\omega\}} (1 - f(\omega)) F_0^d(\{\omega\})^{-1} F_0^d(d\omega) \right)^2 \\
= \sum_{\omega \in \Omega} (1 - f(\omega))^2 \\
= \sum_{\omega \in \Omega} \left( 1 - \frac{F_1^d(\{\omega\})}{F_0^d(\{\omega\})} \right)^2.
\end{aligned}
$$

This proves (iii)

Now suppose (iii) holds. Then $\| \cdot \|_{L^2(F_0)} \asymp \| \cdot \|_{L^2(F_1)}$, so $\| \cdot \|_{L^2(P_0)} \asymp \| \cdot \|_{L^2(P_1)}$ on $\mathrm{span}\{Z(t) : t \in \mathbb{R}^p\}$. Then since $F_0^c = F_1^c$, $f(\omega) = 1$ for $F_0^c$-a.e. $\omega \in \mathbb{R}^p$ and

$$
\begin{aligned}
\|I - K\|^2_{\mathrm{HS}(H_0(\mathbb{R}^p))} &= \|I - M_f\|^2_{\mathrm{HS}(L^2(F_0))} \\
&= \|I - M_f\|^2_{\mathrm{HS}(L^2(\mathbb{R}^p \setminus \Omega, F_0^c))} + \|I - M_f\|^2_{\mathrm{HS}(L^2(\Omega, F_0^d))} \\
&= \|I - M_f\|^2_{\mathrm{HS}(L^2(\Omega, F_0^d))} \\
&= \sum_{\omega \in \Omega} \left(1 - \frac{F_1^d(\{\omega\})}{F_0^d(\{\omega\})}\right)^2 \\
&< \infty.
\end{aligned}
$$

This proves (ii). $\qquad\square$

**We are now in a position to prove Theorem 3.4.**

*Proof of Theorem 3.4.* This is a direct consequence of Lemma B.3 and condition (iii) of Lemma B.4 for the equivalence of measures on $\mathbb{R}^p$. $\qquad\square$

To obtain Theorem 3.5, we want to apply Theorem 3.4 to the families $(K^c)_{\theta \in \Theta}$ and $(K^d)_{\theta \in \Theta}$, but since Theorem 3.4 required the kernels to be holomorphic near 0, we need to show that $K^c$ and $K^d$ are holomorphic near 0 whenever $K$ is.

**Lemma B.5.** *Suppose $K$ is a stationary covariance kernel on $\mathbb{R}^p$ that is holomorphic on the polydisc $D = \{z \in \mathbb{C}^p : |z_j| < \delta_j, \forall j\}$, where $\delta_1, \ldots, \delta_p > 0$. Then both $K^c$ and $K^d$ are holomorphic on $D$.*

*Proof.* Note that $K$ is the characteristic function of $F$. By straightforward generalization of Theorem 2 of Lukacs et al. (1952) from $\mathbb{R}$ to $\mathbb{R}^p$, we have for arbitrary continuous stationary kernel $\widetilde{K}$ that

$$
\widetilde{K} \text{ is holomorphic on } D \iff \int e^{|\omega_1||y_1| + \cdots + |\omega_p||y_p|} \widetilde{F}(d\omega) < \infty \text{ when } |y_j| < \delta_j \text{ for all } j.
$$

Thus $\int e^{|\omega_1||y_1| + \cdots + |\omega_p||y_p|} F(d\omega) < \infty$ for $|y_j| < \delta_j$. Since $F^c(d\omega) \le F(d\omega)$ and $F^d(d\omega) \le F(d\omega)$, it follows that $\int e^{|\omega_1||y_1| + \cdots + |\omega_p||y_p|} F^c(d\omega) < \infty$ and $\int e^{|\omega_1||y_1| + \cdots + |\omega_p||y_p|} F^d(d\omega) < \infty$ for $|y_j| < \delta_j$. Hence $K^c$ and $K^d$ are holomorphic on $D$. $\qquad\square$

**Now we are ready to prove Theorem 3.5.**

*Proof of Theorem 3.5.* 1. Suppose $h(\theta)$ is microergodic for $(K_\theta^c)_{\theta \in \Theta}$ and $g(\theta)$ is microergodic for $(K_\theta^d)_{\theta \in \Theta}$. Let $\theta_1, \theta_2 \in \Theta$. By using Theorem 3.4 to write down the conditions for $K_{\theta_1} \equiv K_{\theta_2}$, $K_{\theta_1}^c \equiv K_{\theta_2}^c$, and $K_{\theta_1}^d \equiv K_{\theta_2}^d$, we obtain

$$
K_{\theta_1} \equiv K_{\theta_2} \iff K_{\theta_1}^c \equiv K_{\theta_2}^c \text{ and } K_{\theta_1}^d \equiv K_{\theta_2}^d.
$$

Then by definition of microergodicity,

$$
\begin{aligned}
K_{\theta_1}^c \equiv K_{\theta_2}^c \text{ and } K_{\theta_1}^d \equiv K_{\theta_2}^d &\iff h(\theta_1) = h(\theta_2) \text{ and } g(\theta_1) = g(\theta_2) \\
&\iff (h(\theta_1), g(\theta_1)) = (h(\theta_2), g(\theta_2)).
\end{aligned}
$$

Thus $K_{\theta_1} \equiv K_{\theta_2} \iff (h(\theta_1), g(\theta_1)) = (h(\theta_2), g(\theta_2))$, so $(h(\theta), g(\theta))$ is microergodic for $(K_\theta)_{\theta \in \Theta}$.

2. Let $\theta_1, \theta_2 \in \Theta$. By Theorem B.5, both $K_{\theta_1}^c$ and $K_{\theta_2}^c$ are holomorphic on some ball around 0. By Theorem 3.4,

$$
K_{\theta_1}^c \equiv K_{\theta_2}^c \iff K_{\theta_1}^c(x) = K_{\theta_2}^c(x) \text{ for all } x \in \mathbb{R}^p.
$$

Similarly, by Theorem B.5, both $K_{\theta_1}^d$ and $K_{\theta_2}^d$ are holomorphic on some ball around 0, and by Theorem 3.4,

$$K_{\theta_1}^d \equiv K_{\theta_2}^d \iff F_{\theta_1}^d(\{\omega\}) \asymp F_{\theta_2}^d(\{\omega\}) \text{ and } \sum_{\omega: F_{\theta_1}^d(\{\omega\}) > 0} \left(1 - \frac{F_{\theta_2}^d(\{\omega\})}{F_{\theta_1}^d(\{\omega\})}\right)^2 < \infty.$$

$\square$

### B.2   Proof of Theorem 3.1

We study the kernels in Table 1 in the order of RBF, Per, Damped Per, RQ and Cosine.

*Proof for SE.* Let $K_{\sigma,\ell}(x)$ denote the SE kernel with parameters $\sigma^2, \ell > 0$. Since the SE kernel has spectral density, by Theorem 3.4, it suffices to show that $K_{\sigma_1,\ell_1}(x) = K_{\sigma_2,\ell_2}(x)$ for all $x$ implies $\sigma_1^2 = \sigma_2^2$ and $\ell_1 = \ell_2$. To show this, it is equivalent to show that $\sigma$ and $\ell$ can be written as functions of $K_{\sigma,\ell}$. We have

$$K_{\sigma,\ell}(x) = \sigma^2 \exp\left(-\frac{\|x\|^2}{2\ell^2}\right),$$

$$\log K_{\sigma,\ell}(x) = \log(\sigma^2) - \frac{\|x\|^2}{2\ell^2},$$

$$\frac{\log K_{\sigma,\ell}(x)}{\|x\|^2} = -\frac{1}{2\ell^2} + O\left(\frac{1}{\|x\|^2}\right) \text{ as } x \to \infty,$$

$$-\frac{1}{2\ell^2} = \lim_{\|x\| \to \infty} \frac{\log K_{\sigma,\ell}(x)}{\|x\|^2},$$

$$\sigma^2 = K_{\sigma,\ell}(0).$$

Thus $\sigma$ and $\ell$ are functions of $K_{\sigma,\ell}$. $\square$

*Proof for Per.* To apply Theorem 3.4, we first need to compute the spectral measures $F_{\sigma,\ell,\gamma}$. The period of $K$ is $\gamma$. Since $K_{1,\ell,2\pi}$ is $2\pi$ periodic, it has Fourier series coefficients $\widehat{K}_{1,\ell,2\pi}(j)$ satisfying

$$K_{1,\ell,2\pi}(x) = \sum_{j=-\infty}^{\infty} \widehat{K}_{1,\ell,2\pi}(j) e^{ijx},$$

$$\widehat{K}_{1,\ell,2\pi}(j) = \frac{1}{2\pi} \int_{-\pi}^{\pi} K_{1,\ell,2\pi}(x) e^{-ijx} dx.$$

Thus

$$K_{\sigma,\ell,\gamma}(x) = \sigma^2 K_{1,\ell,2\pi}\left(\frac{2\pi}{\gamma} x\right)$$

$$= \sigma^2 \sum_{j=-\infty}^{\infty} \widehat{K}_{1,\ell,2\pi}(j) e^{ij\frac{2\pi}{\gamma}x}$$

$$= \sigma^2 \int_{-\infty}^{\infty} \sum_{j=-\infty}^{\infty} \widehat{K}_{1,\ell,2\pi}(j) \delta_{j\frac{2\pi}{\gamma}}(\omega) e^{i\omega x} d\omega.$$

Thus

$$F_{\sigma,\ell,\gamma} = \sigma^2 \sum_{j=-\infty}^{\infty} \widehat{K}_{1,\ell,2\pi}(j) \delta_{j\frac{2\pi}{\gamma}}.$$

By 9.6.19 on page 128 of Abramowitz and Stegun (1964),

$$\widehat{K}_{1,\ell,2\pi}(j) = \frac{1}{2\pi}\int_{-\pi}^{\pi}\exp\left(\frac{-2\sin(x/2)^2}{\ell^2}\right)e^{-ijx}dx$$

$$= \frac{1}{\pi}\int_0^{\pi}\exp\left(\frac{-2\sin(x/2)^2}{\ell^2}\right)\cos(jx)dx$$

$$= \frac{e^{-\frac{1}{\ell^2}}}{\pi}\int_0^{\pi}e^{\frac{\cos(x)}{\ell^2}}\cos(jx)dx$$

$$= \exp\left(-\frac{1}{\ell^2}\right)I_j\left(\frac{1}{\ell^2}\right).$$

By 9.6.10 on page 375 of Abramowitz and Stegun (1964), $\widehat{K}_{1,\ell,2\pi}(j) > 0$ for all $j \in \mathbb{Z}$. By 9.6.3 on page 127 and 9.3.1 on page 117 of Abramowitz and Stegun (1964),

$$I_j\left(\frac{1}{\ell^2}\right) = i^{-j}J_j\left(\frac{i}{\ell^2}\right)$$

$$= i^{-j}\frac{1}{\sqrt{2\pi j}}\left(\frac{e\frac{i}{\ell^2}}{2j}\right)^j(1+O(j^{-1})) \text{ as } j\to\infty$$

$$= \frac{1}{\sqrt{2\pi j}}\left(\frac{e}{2\ell^2 j}\right)^j(1+O(j^{-1})).$$

Thus

$$\widehat{K}_{1,\ell,2\pi}(j) = \exp\left(-\frac{1}{\ell^2}\right)I_j\left(\frac{1}{\ell^2}\right)$$

$$= \frac{\exp\left(-\frac{1}{\ell^2}\right)}{\sqrt{2\pi j}}\left(\frac{e}{2\ell^2 j}\right)^j(1+O(j^{-1}))$$

$$= \frac{1}{\sqrt{2\pi j}}\left(\frac{e}{2j}\right)^j\exp\left(-\frac{1}{\ell^2}\right)\ell^{-2j}(1+O(j^{-1})).$$

Suppose $(\sigma_0, \ell_0, \gamma_0)$ and $(\sigma_1, \ell_1, \gamma_1)$ are two parameter sets with

$$F^d_{\sigma_0,\ell_0,\gamma_0}(\{\omega\}) \asymp F^d_{\sigma_1,\ell_1,\gamma_1}(\{\omega\}),$$

$$\sum_{\omega:F_{\sigma_0,\ell_0,\gamma_0}(\{\omega\})>0}\left(1 - \frac{F^d_{\sigma_1,\ell_1,\gamma_1}(\{\omega\})}{F^d_{\sigma_0,\ell_0,\gamma_0}(\{\omega\})}\right)^2 < \infty.$$

Due to the formula $F_{\sigma,\ell,\gamma} = \sigma^2\sum_{j=-\infty}^{\infty}\widehat{K}_{1,\ell,2\pi}(j)\delta_{j\frac{2\pi}{\gamma}}$ and the fact that $\widehat{K}_{1,\ell,2\pi}(j) > 0$ for all $j$, we must have $\gamma_0 = \gamma_1$ and

$$\sigma_0^2\widehat{K}_{1,\ell_0,2\pi}(j) \asymp \sigma_1^2\widehat{K}_{1,\ell_1,2\pi}(j),$$

$$\sum_{j=-\infty}^{\infty}\left(1 - \frac{\sigma_1^2\widehat{K}_{1,\ell_1,2\pi}(j)}{\sigma_0^2\widehat{K}_{1,\ell_0,2\pi}(j)}\right)^2 < \infty.$$

We have

$$\frac{\sigma_1^2\widehat{K}_{1,\ell_1,2\pi}(j)}{\sigma_0^2\widehat{K}_{1,\ell_0,2\pi}(j)} = \frac{\frac{\sigma_1^2}{\sqrt{2\pi j}}\left(\frac{e}{2j}\right)^j\exp\left(-\frac{1}{\ell_1^2}\right)\ell_1^{-2j}(1+O(j^{-1}))}{\frac{\sigma_0^2}{\sqrt{2\pi j}}\left(\frac{e}{2j}\right)^j\exp\left(-\frac{1}{\ell_0^2}\right)\ell_0^{-2j}(1+O(j^{-1}))}$$

$$= \frac{\sigma_1^2\exp\left(-\frac{1}{\ell_1^2}\right)}{\sigma_0^2\exp\left(-\frac{1}{\ell_0^2}\right)}\left(\frac{\ell_1}{\ell_0}\right)^{-2j}(1+O(j^{-1})).$$

If $\ell_0 \neq \ell_1$, then $\frac{\sigma_1^2 \widehat{K}_{1,\ell_1,2\pi}(j)}{\sigma_0^2 \widehat{K}_{1,\ell_0,2\pi}(j)} \to 0$ or $\infty$ as $j \to \infty$, so we must have $\ell_0 = \ell_1$. Then $\frac{\sigma_1^2 \widehat{K}_{1,\ell_1,2\pi}(j)}{\sigma_0^2 \widehat{K}_{1,\ell_0,2\pi}(j)} = \frac{\sigma_1^2}{\sigma_0^2}(1 + O(j^{-1})) \to \frac{\sigma_1^2}{\sigma_0^2}$, and the assumption $\sum_{j=-\infty}^{\infty} \left(1 - \frac{\sigma_1^2 \widehat{K}_{1,\ell_1,2\pi}(j)}{\sigma_0^2 \widehat{K}_{1,\ell_0,2\pi}(j)}\right)^2 < \infty$ implies $\sigma_0 = \sigma_1$. □

*Proof for Damped Per.* Let $K_{\sigma,\ell_1,\ell_2,\gamma}$ denote the damped periodic kernel with parameters $\sigma, \ell_1, \ell_2, \gamma > 0$. Note that the damped periodic kernel is the product of the SE kernel and periodic kernel, so it's spectral measure is the convolution of the SE spectral measure and the periodic kernel spectral measure. If $f$ denotes the spectral density of the SE kernel, and $\sum_j a_j \delta_{\omega_j}$ denotes the spectral measure of the periodic kernel, then their convolution has spectral density $g(\omega) = \sum_j a_j f(\omega - \omega_j)$. Thus by Theorem 3.4, we need to show that $\sigma, \ell_1, \ell_2, \gamma$ can all be written as functions of $K_{\sigma,\ell_1,\ell_2,\gamma}$. We have

$$K_{\sigma,\ell_1,\ell_2,\gamma}(x) = \sigma^2 \exp\left(-\frac{x^2}{2\ell_1^2} - \frac{2\sin^2\left(\frac{\pi x}{\gamma}\right)}{\ell_2^2}\right).$$

Note that $K_{\sigma,\ell_1,\ell_2,\gamma}$ has a unique holomorphic extension to $\mathbb{C}$ as

$$K_{\sigma,\ell_1,\ell_2,\gamma}(z) = \sigma^2 \exp\left(-\frac{z^2}{2\ell_1^2} - \frac{2\sin^2\left(\frac{\pi z}{\gamma}\right)}{\ell_2^2}\right).$$

Putting $z = iy$ yields

$$K_{\sigma,\ell_1,\ell_2,\gamma}(iy) = \sigma^2 \exp\left(\frac{y^2}{2\ell_1^2} + \frac{2\sinh^2\left(\frac{\pi y}{\gamma}\right)}{\ell_2^2}\right),$$

$$\log K_{\sigma,\ell_1,\ell_2,\gamma}(iy) = \log(\sigma^2) + \frac{y^2}{2\ell_1^2} + \frac{2\sinh^2\left(\frac{\pi y}{\gamma}\right)}{\ell_2^2}.$$

Since $\sinh(x) = \frac{e^x + e^{-x}}{2} \sim \frac{1}{2}e^x$,

$$\log K_{\sigma,\ell_1,\ell_2,\gamma}(iy) \sim \frac{\exp\left(\frac{\pi y}{\gamma}\right)}{2\ell_2^2},$$

$$\log\log K_{\sigma,\ell_1,\ell_2,\gamma}(iy) = \frac{\pi y}{\gamma} - \log(2\ell_2^2) + o(1),$$

$$\frac{\pi}{\gamma} = \frac{\log\log K_{\sigma,\ell_1,\ell_2,\gamma}(iy)}{y} + O(y^{-1}).$$

Thus $\gamma$ is a function of $K_{\sigma,\ell_1,\ell_2,\gamma}$. Then

$$2\ell_2^2 \sim \frac{\exp\left(\frac{\pi y}{\gamma}\right)}{\log K_{\sigma,\ell_1,\ell_2,\gamma}(iy)},$$

so $\ell_2$ is a function of $K_{\sigma,\ell_1,\ell_2,\gamma}$. Note

$$\log K_{\sigma,\ell_1,\ell_2,\gamma}(iy) - \frac{2\sinh^2\left(\frac{\pi y}{\gamma}\right)}{\ell_2^2} = \log(\sigma^2) + \frac{y^2}{2\ell_1^2},$$

$$\frac{1}{2\ell_1^2} = \frac{\log K_{\sigma,\ell_1,\ell_2,\gamma}(iy) - \frac{2\sinh^2\left(\frac{\pi y}{\gamma}\right)}{\ell_2^2}}{y^2} + O(y^{-2}),$$

$$\sigma^2 = K_{\sigma,\ell_1,\ell_2,\gamma}(0).$$

Thus $\ell_1^2$ and $\sigma^2$ are functions of $K_{\sigma,\ell_1,\ell_2,\gamma}$. □

*Proof for RQ.* Let $K_{\sigma,\alpha,\ell}$ denote the RQ kernel with parameters $\sigma^2, \alpha, \ell > 0$. Since the RQ kernel has a spectral density, as before, we need to show that $\sigma^2, \alpha, \ell$ all are functions of $K_{\sigma,\alpha,\ell}$. We have $\sigma^2 = K_{\sigma,\alpha,\ell}(0)$, so $\sigma^2$ is a function of $K_{\sigma,\alpha,\ell}$. Let

$$K(t) = \frac{1}{\sigma^2} K_{\sigma,\alpha,\ell}(te_1)$$
$$= \left(1 + \frac{t^2}{2\alpha\ell^2}\right)^{-\alpha}.$$

Then as $t \to \infty$,

$$K(t) \sim \frac{t^{-2\alpha}}{(2\alpha\ell^2)^{-\alpha}}$$
$$= \frac{2^\alpha \alpha^\alpha \ell^{2\alpha}}{t^{2\alpha}},$$
$$\log K(t) = \log(2^\alpha \alpha^\alpha \ell^{2\alpha}) - 2\alpha \log(t) + o(1),$$
$$\frac{\log K(t)}{\log(t)} = -2\alpha + O\left(\frac{1}{\log(t)}\right).$$

Thus $\alpha$ is a function of $K_{\sigma,\alpha,\ell}$. Since

$$\ell^{2\alpha} \sim K(t)\frac{t^{2\alpha}}{2^\alpha \alpha^\alpha},$$

$\ell$ is a function of $K_{\sigma,\alpha,\ell}$. □

*Proof for Cosine.* Let $K_{\sigma,s}$ denote the cosine kernel with parameters $\sigma^2 > 0$, $s \in \mathbb{R}^p$. The spectral measure is

$$F_{\sigma,s} = \sigma^2 \frac{1}{2}(\delta_{-s} + \delta_s).$$

This is a discrete measure with support $\{-s, s\}$. By Theorem 3.4, $K_{\sigma_1,s_1} \equiv K_{\sigma_2,s_2}$ if and only if

$$F_{\sigma_1,s_1}(\{\omega\}) \asymp F_{\sigma_2,s_2}(\{\omega\}) \text{ and } \sum_{\omega:F_{\sigma_1,s_1}(\{\omega\})>0} \left(1 - \frac{F_{\sigma_2,s_2}(\{\omega\})}{F_{\sigma_1,s_1}(\{\omega\})}\right)^2 < \infty.$$

Since the supports of $F_{\sigma_1,s_1}$ and $F_{\sigma_2,s_2}$ are finite sets, the condition $F_{\sigma_1,s_1}(\{\omega\}) \asymp F_{\sigma_2,s_2}(\{\omega\})$ is equivalent to the condition that the supports are equal, that is, $\{-s_1, s_1\} = \{-s_2, s_2\}$. The second condition always holds because the sum is over a finite set. Thus $K_{\sigma_1,s_1} \equiv K_{\sigma_2,s_2}$ if and only if $\{-s_1, s_1\} = \{-s_2, s_2\}$. Thus $\{-s, s\}$ is microergodic. □

### B.3 Proof of Theorem 3.2

*Proof of Theorem 3.2.* First of all, Theorem 6 on page 123 of Stein (1999) tells us that for continuous kernels $K_1$, $K_2$, the kernels $K_1(x) + \tau_1 1_{\{x=0\}}$ and $K_2(x) + \tau_2 1_{\{x=0\}}$ are equivalent if and only if $\tau_1 = \tau_2$ and $K_1 \equiv K_2$. Thus it suffices to prove identifiability of the parameters under the assumption that $\theta_{11} = 0$.

Write

$$K_{\theta,\gamma}(x) = K_1(x) + K_2(x) + K_3(x) + K_4(x),$$

where

$$K_1(x) = \theta_1^2 \exp\left(-\frac{x^2}{2\theta_2^2}\right),$$

$$K_2(x) = \theta_3^2 \exp\left(-\frac{x^2}{2\theta_4^2} - \frac{2\sin^2\left(\frac{\pi x}{\gamma}\right)}{\theta_5^2}\right),$$

$$K_3(x) = \theta_6^2 \left(1 + \frac{x^2}{2\theta_8\theta_7^2}\right)^{-\theta_8},$$

$$K_4(x) = \theta_9^2 \exp\left(-\frac{x^2}{2\theta_{10}^2}\right).$$

Note that $K_1, K_2, K_3, K_4$ are analytic on the disks of radius $\infty, \infty, (2\theta_8\theta_7^2)^{1/2}, \infty$ respectively. Thus $K_{\theta,\gamma}$ is holomorphic on the disk of radius $(2\theta_8\theta_7^2)^{1/2}$. Since $K_1, K_2, K_3, K_4$ each have a spectral density, $K_{\theta,\gamma}$ has a spectral density, and consequently, $K_{\theta,\gamma}^c = K_{\theta,\gamma}$, $K_{\theta,\gamma}^d = 0$. That is, we focus only on the continuous component.

By Theorem 3.4, we need to show that if $(\theta, \gamma)$ and $(\widetilde{\theta}, \widetilde{\gamma})$ are such that $K_{\theta,\gamma}(x) = K_{\widetilde{\theta},\gamma}(x)$ for all $x \in \mathbb{R}$, then $(\theta, \gamma) = (\widetilde{\theta}, \widetilde{\gamma})$. Let

$$\Theta = \{(\theta, \gamma) : \theta_1, \ldots, \theta_{10}, \gamma > 0, \theta_{10} < \theta_2\},$$
$$\mathcal{C} = \{K_{\theta,\gamma} : (\theta, \gamma) \in \Theta\}.$$

Let $S$ be the collection of functions $h = h(\theta, \gamma)$ with domain $\Theta$ such that there is a function $\varphi : \mathcal{C} \to \Theta$ with $h(\theta, \gamma) = \varphi(K_{\theta,\gamma})$ for all $(\theta, \gamma) \in \Theta$. We need to show that $(\theta, \gamma) \in S$. We have

$$K_{\theta,\gamma}(x) \sim \theta_6^2 (2\theta_8\theta_7^2)^{\theta_8} x^{-2\theta_8},$$
$$\log(K_{\theta,\gamma}(x)) = \log(\theta_6^2(2\theta_8\theta_7^2)^{\theta_8}) - 2\theta_8 \log(x) + o(1),$$
$$\frac{\log(K_{\theta,\gamma}(x))}{\log(x)} = -2\theta_8 + o(\log(x)^{-1}).$$

Thus

$$\theta_8 \in S,$$

and from this we deduce

$$\theta_6^2 \theta_7^{2\theta_8} \in S.$$

Note that the largest $r > 0$ for which $K_{\theta,\gamma}$ extends to a holomorphic function on $D(0, r)$ is $(2\theta_8\theta_7^2)^{1/2}$. Thus $(2\theta_8\theta_7^2)^{1/2} \in S$, so

$$\theta_7 \in S.$$

And from $\theta_6^2 \theta_7^{2\theta_8} \in S$ we obtain

$$\theta_6 \in S.$$

Set

$$f(x) = \theta_1^2 \exp\left(-\frac{x^2}{2\theta_2^2}\right) + \theta_3^2 \exp\left(-\frac{x^2}{2\theta_4^2} - \frac{2\sin^2\left(\frac{\pi x}{\gamma}\right)}{\theta_5^2}\right) + \theta_9^2 \exp\left(-\frac{x^2}{2\theta_{10}^2}\right).$$

Since $\theta_6, \theta_7, \theta_8 \in S$, $f \in S$. Let $f$ denote the unique holomorphic extension of $f$ to $\mathbb{C}$. For $x \in \mathbb{R}$,

$$f(-ix) = \theta_1^2 \exp\left(\frac{x^2}{2\theta_2^2}\right) + \theta_3^2 \exp\left(\frac{x^2}{2\theta_4^2} + \frac{2\sinh^2\left(\frac{\pi x}{\gamma}\right)}{\theta_5^2}\right) + \theta_9^2 \exp\left(\frac{x^2}{2\theta_{10}^2}\right).$$

So as $x \to \infty$,

$$f(-ix) \sim \theta_3^2 \exp\left(\frac{x^2}{2\theta_4^2} + \frac{2\sinh^2\left(\frac{\pi x}{\gamma}\right)}{\theta_5^2}\right),$$

$$\log f(-ix) = \log(\theta_3^2) + \frac{x^2}{2\theta_4^2} + \frac{2\sinh^2\left(\frac{\pi x}{\gamma}\right)}{\theta_5^2} + o(1)$$

$$\sim \frac{2\sinh^2\left(\frac{\pi x}{\gamma}\right)}{\theta_5^2}$$

$$\sim \frac{2}{\theta_5^2}\left(\frac{1}{2}\exp\left(\frac{\pi x}{\gamma}\right)\right)^2$$

$$= \frac{1}{2\theta_5^2}\exp\left(\frac{2\pi x}{\gamma}\right),$$

$$\log\log f(-ix) = -\log(2\theta_5^2) + \frac{2\pi x}{\gamma} + o(1),$$

$$\frac{\log\log f(-ix)}{2\pi x} = \frac{1}{\gamma} + o(x^{-1}).$$

Thus

$$\gamma \in S.$$

Then from $\frac{\log f(-ix)}{\sinh^2\left(\frac{\pi x}{\gamma}\right)} = \frac{2}{\theta_5^2} + o(1)$ we get

$$\theta_5 \in S.$$

We have

$$\log f(-ix) - \frac{2\sinh^2\left(\frac{\pi x}{\gamma}\right)}{\theta_5^2} = \log(\theta_3^2) + \frac{x^2}{2\theta_4^2} + o(1),$$

$$\frac{\log f(-ix) - \frac{2\sinh^2\left(\frac{\pi x}{\gamma}\right)}{\theta_5^2}}{x^2} = \frac{1}{2\theta_4^2} + o(x^{-1}),$$

$$\theta_4 \in S,$$

$$\log f(-ix) - \frac{2\sinh^2\left(\frac{\pi x}{\gamma}\right)}{\theta_5^2} - \frac{x^2}{2\theta_4^2} = \log(\theta_3^2) + o(1),$$

$$\theta_3 \in S.$$

Let

$$g(x) = f(-ix) - \theta_3^2 \exp\left(\frac{x^2}{2\theta_4^2} + \frac{2\sinh^2\left(\frac{\pi x}{\gamma}\right)}{\theta_5^2}\right)$$

$$= \theta_1^2 \exp\left(\frac{x^2}{2\theta_2^2}\right) + \theta_9^2 \exp\left(\frac{x^2}{2\theta_{10}^2}\right).$$

Since $\theta_2 < \theta_{10}$,

$$g(x) \sim \theta_1^2 \exp\left(\frac{x^2}{2\theta_2^2}\right),$$

$$\log(g(x)) = \log(\theta_1^2) + \frac{x^2}{2\theta_2^2},$$

$$\theta_2 \in S,$$
$$\theta_1 \in S.$$

Then by applying the same argument to $g(x) - \theta_1^2 \exp\left(\frac{x^2}{2\theta_2^2}\right)$, we get $\theta_9^2 \exp\left(\frac{x^2}{2\theta_{10}^2}\right) \in S.$ $\quad\square$

## B.4 Proof of Theorems 3.6 to 3.9

*Proof of Theorem 3.6.* Note that $K_{\sigma,M} = \sigma^2 K(M^{1/2}x)$, where $K(x) = \exp\left(-\frac{1}{2}\|x\|^2\right)$ is the standard squared exponential kernel. Let $f = \widehat{K}$. Let $A = M^{1/2}$. Since

$$
K(Ax) = \int f(\omega)e^{i\omega^T Ax}\, d\omega
$$
$$
= \int f(A^{-T}y)e^{iy^T x}\frac{1}{|\det A|}\, dy,
$$

we obtain

$$
\widehat{K_{\sigma,M}}(\omega) = \sigma^2 \frac{1}{(\det M)^{1/2}} f(M^{-1/2}\omega).
$$

Thus $K_{\sigma,M}$ has a spectral density. Suppose $K_{\sigma_0,M_0} \equiv K_{\sigma_1,M_1}$. By Theorem 3.4, $\sigma_0^2 \exp\left(-\frac{1}{2}x^T M_0 x\right) = \sigma_1^2 \exp\left(-\frac{1}{2}x^T M_1 x\right)$ for all $x \in \mathbb{R}^p$. Putting $x = 0$ yields $\sigma_0^2 = \sigma_1^2$, and then taking logs yields $x^T M_0 x = x^T M_1 x$ for all $x \in \mathbb{R}^p$. Since $M_0$ and $M_1$ are symmetric, the polarization identity yields $M_0 = M_1$. Thus $(\sigma^2, M)$ is identifiable, as claimed. □

*Proof of Theorem 3.7.* The spectral measure is

$$
F_{\sigma_1,\ldots,\sigma_m,s_1,\ldots,s_m} = \sum_{j=1}^m \frac{\sigma_j^2}{2}(\delta_{-s_j} + \delta_{s_j}).
$$

The support of $F_{\sigma_1,\ldots,\sigma_m,s_1,\ldots,s_m}$ is $\{\pm s_j : j = 1,\ldots,m\}$. As in the proof of for the cosine kernel, we conclude that $\{\pm s_j : j = 1,\ldots,m\}$ is microergodic. Since we assumed $0 \le s_1 < \cdots < s_m$, the largest $m$ elements of the set $\{\pm s_j : j = 1,\ldots,m\}$ are $s_m, s_{m-1},\ldots,s_1$. Hence $(s_1,\ldots,s_m)$ is microergodic. □

*Proof of Theorem 3.8.* If $K_1$ and $K_2$ are stationary kernels with $K_1(0) = K_2(0) = 1$, then the spectral measure of $K_1 K_2$ is the convolution $F_1 * F_2$, which is defined as the distribution of $X_1 + X_2$ when $X_j \sim F_j$ and $X_1$ and $X_2$ are independent. Thus the support of $F_{\sigma,s_1,\ldots,s_m}$ is $\{\pm s_1 \pm s_2 \pm \cdots \pm s_m\}$. As in the proof for the cosine kernel, we conclude that $\{\pm s_1 \pm s_2 \pm \cdots \pm s_m\}$ is microergodic.

Now suppose $m \in \{1, 2, 3\}$, say $m = 3$. Then the 3 largest elements of the set $\{\pm s_1 \pm s_2 \pm s_3\}$ are $s_1 + s_2 + s_3$, $-s_1 + s_2 + s_3$, and $s_1 - s_2 + s_3$. From these we can algebraically solve for $s_1, s_2, s_3$. This shows that $(s_1, s_2, s_3)$ is a function of $\{\pm s_1 \pm s_2 \pm s_3\}$. Conversely, $\{\pm s_1 \pm s_2 \pm s_3\}$ is also a function of $(s_1, s_2, s_3)$. Thus $(s_1, s_2, s_3)$ is microergodic. □

*Proof of Theorem 3.9.* This proof mimics the proof for the sum of cosine kernels, but is a bit more elaborate due to the infinite support of the spectral measures. Let $\widehat{K}_\ell(j)$, $j \in \mathbb{Z}$ denote the Fourier series coefficients of the periodic kernel with $\sigma^2 = 1$, length-scale $\ell$, and $\gamma = 2\pi$. In the proof for the periodic kernel, we showed that

$$
F_{\sigma,\ell,\gamma} = \sum_{j=-\infty}^{\infty} \sigma^2 \widehat{K}_\ell(j)\delta_{j\frac{2\pi}{\gamma}},
$$

$$
\widehat{K}_\ell(j) > 0 \text{ for all } j \in \mathbb{Z},
$$

$$
\widehat{K}_\ell(j) = \frac{1}{\sqrt{2\pi j}}\left(\frac{e}{2j}\right)^j \exp\left(-\frac{1}{\ell^2}\right)\ell^{-2j}(1 + O(j^{-1})), \tag{3}
$$

$$
\frac{\sigma_1^2 \widehat{K}_{\ell_1}(j)}{\sigma_0^2 \widehat{K}_{\ell_0}(j)} = \frac{\sigma_1^2 \exp\left(-\frac{1}{\ell_1^2}\right)}{\sigma_0^2 \exp\left(-\frac{1}{\ell_0^2}\right)}\left(\frac{\ell_1}{\ell_0}\right)^{-2j}(1 + O(j^{-1})). \tag{4}
$$

Suppose $K_{\sigma_1,\ell_1,\gamma_1,\sigma_2,\ell_2,\gamma_2} \equiv K_{\widetilde{\sigma}_1,\widetilde{\ell}_1,\widetilde{\gamma}_1,\widetilde{\sigma}_2,\widetilde{\ell}_2,\widetilde{\gamma}_2}$. Write

$$s_1 = \frac{2\pi}{\gamma_1},$$
$$s_2 = \frac{2\pi}{\gamma_2},$$
$$\widetilde{s}_1 = \frac{2\pi}{\widetilde{\gamma}_1},$$
$$\widetilde{s}_2 = \frac{2\pi}{\widetilde{\gamma}_2}.$$

Note that $0 < s_1 < s_2$ and $0 < \widetilde{s}_1 < \widetilde{s}_2$. Let $F$ and $\widetilde{F}$ denote the spectral measures of $K_{\sigma_1,\ell_1,\gamma_1,\sigma_2,\ell_2,\gamma_2}$ and $K_{\widetilde{\sigma}_1,\widetilde{\ell}_1,\widetilde{\gamma}_1,\widetilde{\sigma}_2,\widetilde{\ell}_2,\widetilde{\gamma}_2}$. We have

$$F = \sum_{j=-\infty}^{\infty} \sigma_1^2 \widehat{K}_{\ell_1}(j)\delta_{js_1} + \sum_{j=-\infty}^{\infty} \sigma_2^2 \widehat{K}_{\ell_2}(j)\delta_{js_2},$$

$$\widetilde{F} = \sum_{j=-\infty}^{\infty} \widetilde{\sigma}_1^2 \widehat{K}_{\widetilde{\ell}_1}(j)\delta_{j\widetilde{s}_1} + \sum_{j=-\infty}^{\infty} \widetilde{\sigma}_2^2 \widehat{K}_{\widetilde{\ell}_2}(j)\delta_{j\widetilde{s}_2}.$$

By Theorem 3.4, the supports of $F$ and $\widetilde{F}$ are the same, that is,

$$\{js_1 : j \in \mathbb{Z}\} \cup \{js_2 : j \in \mathbb{Z}\} = \{j\widetilde{s}_1 : j \in \mathbb{Z}\} \cup \{j\widetilde{s}_2 : j \in \mathbb{Z}\}.$$

Equating the smallest positive element of both sides yields

$$s_1 = \widetilde{s}_1.$$

Now split the rest of the proof into two cases. Case 1: Suppose $\frac{s_2}{s_1} \notin \mathbb{Q}$. Since $s_2 \in \operatorname{supp}(\widetilde{F})$ and $s_2$ is not an integer multiple of $s_1$, there is $j \in \mathbb{Z}$ such that $s_2 = j\widetilde{s}_2$. This implies that $\widetilde{s}_2$ cannot be an integer multiple of $s_1$, and since $\widetilde{s}_2 \in \operatorname{supp}(F)$ there must exist $k \in \mathbb{Z}$ such that $\widetilde{s}_2 = ks_2 = jk\widetilde{s}_2$. Thus $jk = 1$, and therefore $j = k = 1$. Thus $s_2 = \widetilde{s}_2$. Now note that

$$\operatorname{supp}(F) \cap (0,\infty) = \{js_1 : j \in \mathbb{N}\} \cup \{js_2 : j \in \mathbb{N}\},$$

and since $\frac{s_2}{s_1} \notin \mathbb{Q}$, the sets $\{js_1 : j \in \mathbb{N}\}$ and $\{js_2 : j \in \mathbb{N}\}$ are disjoint. By Theorem 3.4 and the symmetry $\widehat{K}_\ell(j) = \widehat{K}_\ell(-j)$,

$$\infty > \sum_{\omega:F(\omega)>0} \left(1 - \frac{F(\omega)}{\widetilde{F}(\omega)}\right)^2$$

$$= C + 2\sum_{j=1}^{\infty} \left(1 - \frac{\widetilde{\sigma}_1^2 \widehat{K}_{\widetilde{\ell}_1}(j)}{\sigma_1^2 \widehat{K}_{\ell_1}(j)}\right)^2 + 2\sum_{j=1}^{\infty} \left(1 - \frac{\widetilde{\sigma}_2^2 \widehat{K}_{\widetilde{\ell}_2}(j)}{\sigma_2^2 \widehat{K}_{\ell_2}(j)}\right)^2.$$

In particular, $\frac{\widetilde{\sigma}_1^2 \widehat{K}_{\widetilde{\ell}_1}(j)}{\sigma_1^2 \widehat{K}_{\ell_1}(j)} \to 1$ and $\frac{\widetilde{\sigma}_2^2 \widehat{K}_{\widetilde{\ell}_2}(j)}{\sigma_2^2 \widehat{K}_{\ell_2}(j)} \to 1$ as $j \to \infty$. From (4), we conclude $\ell_1 = \widetilde{\ell}_1$, $\sigma_1 = \widetilde{\sigma}_1$, $\ell_2 = \widetilde{\ell}_2$, and $\sigma_2 = \widetilde{\sigma}_2$.

Case 2: $\frac{s_2}{s_1} \in \mathbb{Q}$. Since $\widetilde{s}_2$ is an integer multiple of $s_1$ or $s_2$, $\frac{\widetilde{s}_2}{s_1} \in \mathbb{Q}$. By writing $s_2 = qs_1$, $\widetilde{s}_2 = \widetilde{q}s_1$ with $q, \widetilde{q} \in \mathbb{Q}$, it is clear that $m = \operatorname{lcm}(s_1, s_2, \widetilde{s}_2) = \operatorname{lcm}(s_1, qs_1, \widetilde{q}s_1)$ exists. Note that

$$F(jm) = \sigma_1^2 \widehat{K}_{\ell_1}\left(j\frac{m}{s_1}\right) + \sigma_2^2 \widehat{K}_{\ell_2}\left(j\frac{m}{s_2}\right).$$

Equation (3) can be written as $\widehat{K}_\ell(j) = j^{-1/2}j^{-j}C(\ell)^j$, where $C(\ell) > 0$. Thus when $\alpha > 1$ is an integer,

$$\frac{\widehat{K}_{\ell_1}(\alpha j)}{\widehat{K}_{\ell_2}(j)} \sim \frac{\alpha^{-1/2}j^{-1/2}\alpha^{-\alpha j}j^{-\alpha j}C(\ell_1)^{\alpha j}}{j^{-1/2}j^{-j}C(\ell_2)^j}$$

$$= \alpha^{-1/2}j^{(1-\alpha)j}C(\alpha, \ell_1, \ell_2)^j,$$

$$\log \frac{\widehat{K}_{\ell_1}(\alpha j)}{\widehat{K}_{\ell_2}(j)} = (1-\alpha)j\log j + j\log C(\alpha, \ell_1, \ell_2) + O(1)$$

$$\to -\infty \text{ as } j \to \infty,$$

so

$$\frac{\widehat{K}_{\ell_1}(\alpha j)}{\widehat{K}_{\ell_2}(j)} \to 0. \tag{5}$$

Therefore, since $\frac{m}{s_2} < \frac{m}{s_1}$,

$$F(jm) = \sigma_1^2 \widehat{K}_{\ell_1}\left(j\frac{m}{s_1}\right) + \sigma_2^2 \widehat{K}_{\ell_2}\left(j\frac{m}{s_2}\right)$$

$$\sim \sigma_2^2 \widehat{K}_{\ell_2}\left(j\frac{m}{s_2}\right).$$

For the same reason,

$$\widetilde{F}(jm) \sim \widetilde{\sigma}_2^2 \widehat{K}_{\widetilde{\ell}_2}\left(j\frac{m}{\widetilde{s}_2}\right).$$

Since $\infty > \sum_{j=1}^\infty \left(1 - \frac{\widetilde{F}(jm)}{F(jm)}\right)^2$, we have $\frac{\widetilde{F}(jm)}{F(jm)} \to 1$, so $\frac{\widetilde{\sigma}_2^2 \widehat{K}_{\widetilde{\ell}_2}\left(j\frac{m}{\widetilde{s}_2}\right)}{\sigma_2^2 \widehat{K}_{\ell_2}\left(j\frac{m}{s_2}\right)} \to 1$. Again by (5), we conclude that $\widetilde{s}_2 = s_2$, and by (4), we obtain $\widetilde{\ell}_2 = \ell_2$ and $\widetilde{\sigma}_2^2 = \sigma_2^2$. Recall $m = \text{lcm}(s_1, s_2) \geq s_2 > s_1$. Thus $s_1, \left(1 + \frac{m}{s_1}\right)s_1, \left(1 + 2\frac{m}{s_1}\right)s_1, \ldots$ are all not integer multiples of $s_2$. Thus

$$F\left(\left(1 + j\frac{m}{s_1}\right)s_1\right) = \sigma_1^2 \widehat{K}_{\ell_1}\left(1 + j\frac{m}{s_1}\right),$$

$$\widetilde{F}\left(\left(1 + j\frac{m}{s_1}\right)s_1\right) = \widetilde{\sigma}_1^2 \widehat{K}_{\widetilde{\ell}_1}\left(1 + j\frac{m}{s_1}\right).$$

Since $\frac{\widetilde{F}\left(\left(1 + j\frac{m}{s_1}\right)s_1\right)}{F\left(\left(1 + j\frac{m}{s_1}\right)s_1\right)} \to 1$, we have $1 = \lim_{j \to \infty} \frac{\widetilde{\sigma}_1^2 \widehat{K}_{\widetilde{\ell}_1}\left(1 + j\frac{m}{s_1}\right)}{\sigma_1^2 \widehat{K}_{\ell_1}\left(1 + j\frac{m}{s_1}\right)} = \lim_{\nu \to \infty} \frac{\widetilde{\sigma}_1^2 \widehat{K}_{\widetilde{\ell}_1}(\nu)}{\sigma_1^2 \widehat{K}_{\ell_1}(\nu)}$. From (4), we deduce $\widetilde{\ell}_1 = \ell_1$ and $\widetilde{\sigma}_1^2 = \sigma_1^2$. $\qquad\square$

### B.5 AN ADDITIONAL THEOREM

In this subsection, we introduce a theorem that is useful in conjunction with Theorem 3.4 for determining the identifiable functions in new kernels constructed from existing ones.

Let $\mathcal{C}$ be the set of continuous stationary kernels $K$ on $\mathbb{R}^p$ for which $K = K^c$, and let $\mathcal{D}$ be the set of continuous stationary kernels for which $K = K^d$. Theorem 3.4 tells us $K_1 \equiv K_2$ if and only if $K_1^c = K_2^c$ and $K_1^d \equiv K_2^d$. Thus it can help to know how to decompose $K$ into $K^c$ and $K^d$. When $K$ is a sum of products of other fully discrete or fully continuous spectrum kernels, the decomposition is easily found by inspection:

**Theorem B.6.** *Suppose $K_0$ and $K_1$ are continuous stationary kernels on $\mathbb{R}^p$ with spectral measures $\widehat{K}_0 = F_0$ and $\widehat{K}_1 = F_1$. The following hold*

*(i)* $\widehat{K_0 + K_1} = F_0 + F_1$

*(ii)* $\widehat{K_0 K_1} = F_0 * F_1$.

*Consequently,*

*(iii)* $K_0 \in \mathcal{C}, K_1 \in \mathcal{C} \implies K_0 + K_1 \in \mathcal{C}$,

*(iv)* $K_0 \in \mathcal{C} \implies K_0 K_1 \in \mathcal{C}$,

*(v)* $K_0 \in \mathcal{D}, K_1 \in \mathcal{D} \implies K_0 + K_1 \in \mathcal{D}, K_0 K_1 \in \mathcal{D}$.

*Proof.* (i) and (ii) are standard results for Fourier transforms. (iii) holds since the sum of two continuous measures is continuous. (iv) holds since the convolution of a continuous measure with any other measure is continuous. (v) holds since the sum of two discrete measures is discrete and the convolution of two discrete measures is discrete. □

This theorem, together with Theorem 3.4, allows us to study the identifiability of parameters in new kernels constructed as the multiplication and summation of existing kernels.

## C    RELATION BETWEEN PRAMETER INFERENCE AND PREDICTION

While the primary focus of this paper is on parameter identifiability in GPs models, it is worth briefly discussing its connection to prediction performance, as prediction is one of the most common applications of GPs.

In the literature, there is a well-established distinction between parameter inference and prediction accuracy. For example, as shown in Theorem 8 of Chapter 4 in Stein (1999), if two measures $P_1$ and $P_0$ are equivalent, then assuming $P_0$ is the true measure and using $P_1$ to obtain the best linear predictor $e_1$ at a new observation location $x_0$, the ratio of the MSE of $e_1$ to the MSE of the best linear predictor $e_0$ under $P_0$ converges to 1 as the sample size $n \to \infty$.

For the Matérn family with known smoothness parameter $\nu$, Theorem 12 of Chapter 4 in Stein (1999) further shows that the asymptotic ratio of the MSEs under two Matérn kernels parameterized by $(\sigma_1^2, \ell_1)$ and $(\sigma_2^2, \ell_2)$ converges to 1, regardless of the values of the parameters (Equation 49, $c = \frac{\sigma_1^2 \ell_2^{2\nu}}{\sigma_2^2 \ell_1^{2\nu}}$). As a consequence, we get asymptotically optimal prediction performance by having $P_1$ be in the correct parametric family, even if $P_1 \perp P_0$. This underscores an important point: prediction is, in an informal sense, "simpler" than parameter inference, as incorrect parameter specification or parameter estimates may still yield asymptotically optimal predictions.

Beyond Matérn kernels, for the holomorphic kernels studied in this paper, Stein's Theorem 8 still holds, However, this is beyond the scope of the current paper, which primarily focuses on parameter inference.

## D    ADDITIONAL FIGURES

In Figure 3, we present the simulations repeated for $\varepsilon = 0.1$ side by side with $\varepsilon = 0.01$. It appears that the variances of the MLEs are smaller when $\varepsilon = 0.01$.

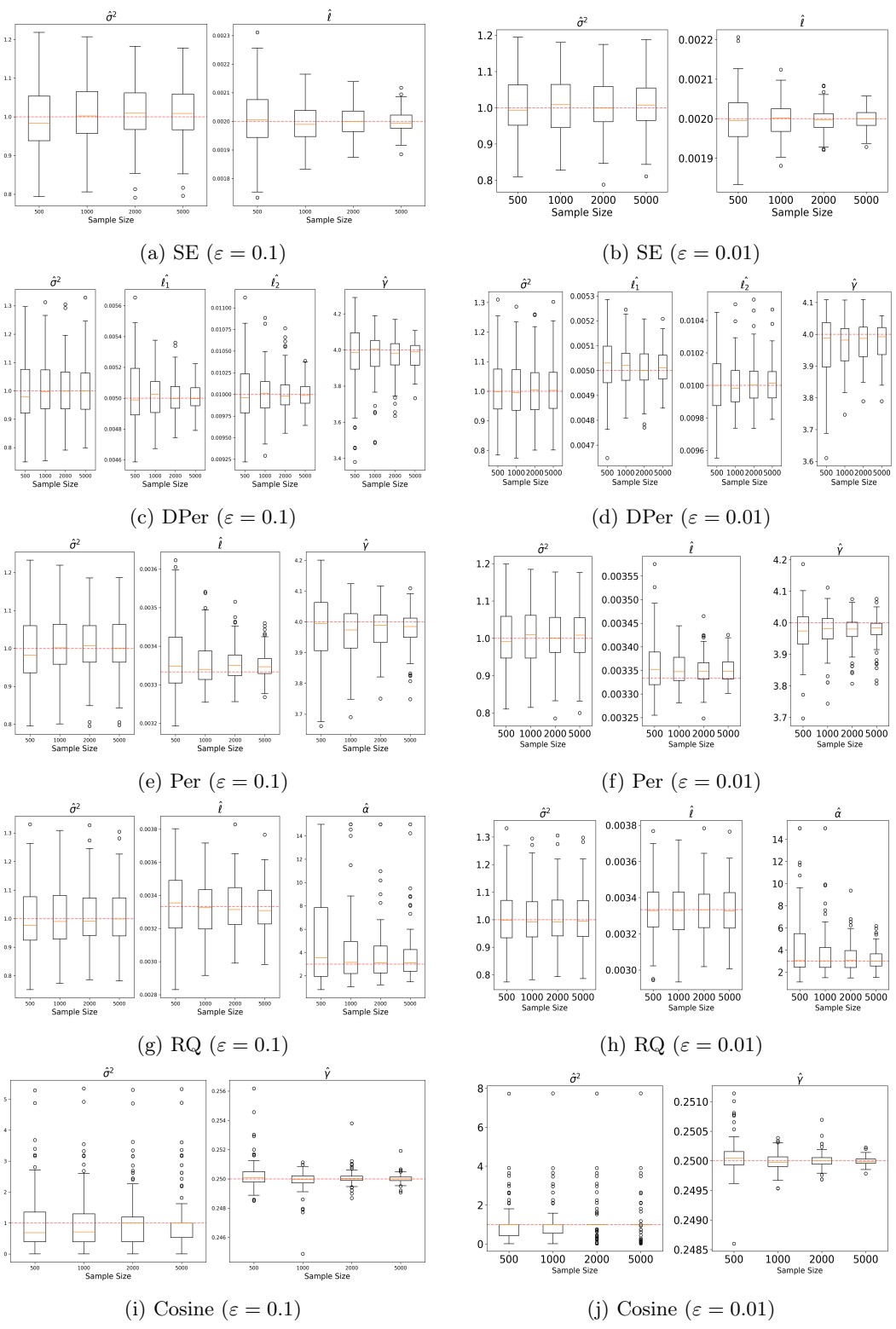

Figure 3: Simulation results for various kernel types with $\varepsilon = 0.1$ and $\varepsilon = 0.01$. Each subfigure shows the boxplots of MLEs for the corresponding kernel, with ground truth in horizontal dashed line.

## E    ADDITIONAL EXPERIMENTAL DETAILS

The ground truth parameters used in Section 4.2 are given by the following table:

| $\theta_1$ | $\theta_2$ | $\theta_3$ | $\theta_4$ | $\theta_5$ | $\gamma$ | $\theta_6$ | $\theta_7$ | $\theta_8$ | $\theta_9$ | $\theta_{10}$ |
|------|------|------|------|------|---|-------|------|------|-------|-------|
| 44.8 | 51.6 | 2.64 | 91.5 | 1.48 | 1 | 0.536 | 2.89 | 8.97 | 0.188 | 0.122 |

Table 3: Ground truth parameters for the combined kernel.

All the experiments were run on a Linux-based virtual computer with 6500 conventional compute cores 472 delivering 13,000 threads. We used 24 CPUs.

Table 4: Runtime and memory usage for each experiment

| Experiment | Time | Memory |
|------------|------|--------|
| Squared Exponential $\varepsilon = 0.1$ | 8 hours, 53 minutes | 32 GB |
| Damped Periodic $\varepsilon = 0.1$ | 6 hours, 24 minutes | 32 GB |
| Periodic $\varepsilon = 0.1$ | 10 hours, 45 minutes | 32 GB |
| Rational Quadratic $\varepsilon = 0.1$ | 10 hours, 16 minutes | 32 GB |
| Cosine $\varepsilon = 0.1$ | 7 hours, 49 minutes | 32 GB |
| Squared Exponential $\varepsilon = 0.01$ | 7 hours, 30 minutes | 32 GB |
| Damped Periodic $\varepsilon = 0.01$ | 6 hours, 45 minutes | 32 GB |
| Periodic $\varepsilon = 0.01$ | 10 hours, 12 minutes | 32 GB |
| Rational Quadratic $\varepsilon = 0.01$ | 9 hours, 42 minutes | 32 GB |
| Cosine $\varepsilon = 0.01$ | 5 hours, 44 minutes | 32 GB |
| Combined Kernel | 1 hour, 21 minutes | 32 GB |

Acknowledgment of Assets Used:

Code Libraries: We utilized the following Python libraries in our program:

- NumPy: BSD License
- SciPy: BSD 3-Clause "New" or "Revised" License
- Matplotlib: PSF License Agreement for Python
- Scikit-learn: BSD 3-Clause "New" or "Revised" License

