# OpenReview forum: "Identifiability for Gaussian Processes with Holomorphic Kernels"
_ICLR.cc/2025/Conference — ICLR 2025 Poster_

### Official Review · Reviewer_iSg8 · 2024-10-17

**Soundness:** 3
**Presentation:** 3
**Contribution:** 3
**Rating:** 8
**Confidence:** 4

**Summary:**

This paper addresses the identifiability issue of the hyperparameters of Gaussian Processes when holomorphic kernels are used. A novel theory is proposed that may help identify when hyperparameters are identifiable and when not, allowing meaningful interpretations in the future. From my own experience, the identifiability of hyperparameters for GPs is an important issue to address.

The writing style and obvious practical implications made this paper an easy and interesting read.

**Strengths:**

- well written.
- a great contribution to answering the long-standing and practical-important question of identifiability for GPs.
- digestible theory.
- immediate practical impact.

**Weaknesses:**

This is a strong paper, I only have one major and a few minor issues with some of the presentation:
Major:
(1) The simulation results, while certainly confirming the theory, are insufficient to convince a critic. Identifiability of hyperparameters is such a rich and practical topic that the results seem somewhat bleak and unimpressive. My suggestions are to apply the methodology to a real dataset --- something of high impact, such as climate, or the mentioned spatial transcriptomics --- show the effects of misidentification by plotting and evaluating results (CRPS, RMSE) and possibly showing relevant snapshots of the log marginal likelihood function. In addition, where the data comes from right now is not well explained. The inputs are defined, but it only says "After generating the outputs". Please be more specific about where the data comes from.

Minor:
(1) There are some errors in the text "summarize(s) existing literature", or table "above" when it is below.
(2) The paragraph after Def. 3 is a little convoluted, possibly due to a grammar mistake, but it's hard to tell. Please be a little more specific and explicit in your logic there.
(3) It would be valuable to the reader to discuss how other training methods (MCMC, Variation Inference) deal with the identifyability challenge. I would guess MCMC as a sampler might deal really well. This would be a great addition.

**Questions:**

- What is the impact of identifiability on the function approximation and uncertainty quantification of a real dataset?
- How does non-identifiability might affect prediction quality?
- What does the log marginal likelihood function look like in such a case?
- How do MCMC and variational inference deal with non-identifyability?

---

> ### Author Response · Authors · 2024-11-18
>
> Thank you for the positive score and for recognizing the significance and practical relevance of our work. Below, we provide a point-by-point response to your comments and questions, and we have revised the paper accordingly, with changes highlighted in magenta.
>
> **Real Data**
>
> Thank you for raising the question about applying our methodology to real datasets. We would like to clarify the scope and intent of our simulation results.
>
> Identifiability is fundamentally a property of parameters in a kernel within a known parametric family. To test this, we generate data from a GP within the parametric family of interest, where the ground truth for the kernel parameters is known. This setup allows us to evaluate whether the MLE or other estimators behave in accordance with the identifiability properties established in the paper.
>
> For real datasets, the true kernel parameters and even the correct parametric family are typically unknown. As a result, directly validating identifiability theory on such data is not feasible. Real datasets are more suitable for evaluating prediction performance and other downstream metrics, rather than studying identifiability directly. We address this consideration in the following discussion on prediction performance.
>
> **Function Approximation/Prediction Quality**
>
> Thank you for this excellent question regarding the relationship between parameter identifiability and prediction performance. We give a detailed explanation in the new Prediction section in Appendix C. We briefly summarize it here.
>
>  In the literature, there is a well-established distinction between parameter inference and prediction accuracy. For example, as shown in Theorem 8 of Chapter 4 in Stein (1999), if two measures $P_1$ and $P_0$ are equivalent, then assuming $P_0$ is the true measure and using $P_1$ to obtain the best linear predictor $e_1$ at a new observation location $x_0$, the ratio of the MSE of $e_1$ to the MSE of the best linear predictor $e_0$ under $P_0$ converges to $1$ as the sample size $n\to\infty$.
>
> For the Matérn family with known smoothness parameter $\nu$, Theorem 12 of Chapter 4 in Stein (1999) further shows that the asymptotic ratio of the MSEs under two Matérn kernels parameterized by $(\sigma_1^2,\ell_1)$ and $(\sigma_2^2,\ell_2)$ converges to 1, regardless of the values of the parameters (Equation 49, $c=\frac{\sigma_1^2\ell_2^{2\nu}}{\sigma_2^2\ell_1^{2\nu}}$). As a consequence, we get asymptotically optimal prediction performance by having $P_1$ be in the correct parametric family, even if $P_1 \perp P_0$. This underscores an important point: prediction is, in an informal sense, “simpler” than parameter inference, as incorrect parameter specification or parameter estimates may still yield asymptotically optimal predictions.
>
> Beyond Matérn kernels, for the holomorphic kernels studied in our paper, Stein's Theorem 8 still holds, ensuring asymptotic equivalence of MSEs for equivalent measures. However, since prediction performance is not the primary focus of our work, we did not include this discussion in the main part of the manuscript. Thank you for raising this point.
>
> **Log-Marginal Likelihood**
>
> The behavior of the maximum likelihood estimators (MLEs) and the log-likelihood function for non-identifiable parameters is an intriguing open problem without a general resolution. For well-studied kernels like the Matérn kernel, even under controlled conditions, the behavior remains complex and poorly understood. For example, consider the Matérn kernel on $[0,1]^2$ with a fixed smoothness $\nu=1/2$. In this case, neither $\sigma^2$ nor $\ell$ is identifiable, but $\sigma^2/\ell$ (Zhang 2004). Specifically, the third row of Figure 3 in Tang et al (2021) illustrates a flat region in the log-likelihood, implying the absence of a clear unique maximizer.
>
> While we recognize the importance of studying the behavior of MLEs and the log-likelihood function, these are fundamentally different from our focus on identifiability. Studying MLE behavior often involves analyzing the likelihood function, whereas identifiability targets the equivalence of Gaussian process laws, employing distinct techniques. As discussed in Section 5, we view the exploration of MLE behavior as an interesting and challenging avenue for future research.

---

> ### Author Response · Authors · 2024-11-18
> **Official Comment by Authors (Continued)**
>
> **Other Potential Estimators such as MCMC and VI**
>
> If a parameter is not identifiable, **no** consistent estimator exists—this holds regardless of the method used, whether it is MLE, MCMC, or VI. However, when parameters are identifiable, consistent estimators **might exist**. In such cases, studying whether MLE, MCMC, or VI outputs a consistent estimator—or designing a new consistent estimator—becomes an interesting and challenging problem. For instance, Loh and Sun (2023) designed a consistent estimator for identifiable parameters in the Matérn kernel, introducing a method distinct from MLE, MCMC, and VI. Such designs are highly case-specific and depend on the properties of the kernel.
>
> Designing consistent estimators for kernels with holomorphic properties is an exciting direction for future work. We appreciate this question, as it aligns with our broader interest in tackling such case-by-case challenges in GP parameter inference.
>
> **Minor Comments**
> Thank you for the detailed and careful check. We have addressed these points in the revised paper.
>
> Thanks again for the helpful comments, which have improved the paper. We remain open to further discussion or clarification.
>
>
> **Reference**
>
> M Stein, Interpolation of spatial data: some theory for kriging, 1999.
>
> H Zhang, Inconsistent estimation and asymptotically equal interpolations in model-based geostatistics, Journal of the American Statistical Association, 2004.
>
> CG Kaufman, BA Shaby, The role of the range parameter for estimation and prediction in geostatistics, Biometrika, 2013.
>
> WL Loh and S Sun, Estimating the parameters of some common Gaussian random fields with nugget under fixed-domain asymptotics, Bernoulli, 2023.

---

> > ### Comment · Reviewer_iSg8 · 2024-11-28
> >
> > Thank you for your response. I believe my score is reflective of the quality of the manuscript.

---

### Official Review · Reviewer_Ec9P · 2024-11-03

**Soundness:** 3
**Presentation:** 2
**Contribution:** 2
**Rating:** 6
**Confidence:** 3

**Summary:**

The paper studies identifiability of the parameters of certain kernels (squared exponential, periodic, rational quadratic, and cosine) and some of their sums and products, in Gaussian process regression (fixed domain asymptotic scenario). In certain cases, it finds microergodic parameters: the functions of parameters that _are_ identifiable, and thus can be consistently estimated. It illustrates the results with an empirical study of convergence of maximum likelihood estimators.

**Strengths:**

- The paper presents new theoretical results on identifiability, which are, in principle, intersting.
- The results look plausible (I didn’t check the 13 pages of proofs in the appendix though).
- The code is provided in the supplementary materials.

**Weaknesses:**

- The paper promises a theoretical framework for determining the identifiability of parameters in arbitrary combinations of squared exponential, periodic, rational quadratic, and cosine kernels. However, the results don’t appear as such: they consider a few special cases, like sums and products of cosine kernels only. The general sum/product combination of the mentioned kernels is not explicitly handled.
- The proofs are hidden in appendices. This is normal for a theoretical paper submitted to a conference. However, in such cases I expect the main ideas/methods to be covered in the main text, so that a reader who doesn’t have time to read the actual proofs could at least get some idea of how they go and how plausible they are.
I believe the paper falls short of this. In particular, the paper studies kernels on $\mathbb{R}^n$, but uses the term “holomorphic” from the world of complex numbers, and assumes the properties of kernels with traditionally real inputs on complex domains. This connection between the world of real-input kernels and the world of complex-analytic methods seems crucial, but remains unclear to me from the main text.
- There are certain problems with writing/presentation, like tables going over the right margin (Table 2) and figures with overly small fonts (Figure 1).
- Empirical study doesn’t illustrate the theoretical findings well. It doesn’t identify them as wrong, but in many of the studied cases does not really show anything.
- I noticed that in the second paragraph of Introduction, all the citations are copied from the 4th paragraph  of introduction of https://proceedings.neurips.cc/paper_files/paper/2023/file/dea2b4f9012686bcc1f59a62bcd28158-Paper-Conference.pdf. The general flow is also similar. The mentioned paper seems to tackle a different problem, so I don't suspect this paper plagiarizing it, but borrowings like this don’t look good. I hope the authors will make sure not to copy-paste things (almost) directly from other papers.

**Questions:**

A key technique used with RBF kernels (and not only with them) is automatic relevance determination (ARD), when each input coordinate corresponds to its own length scale parameter, each optimizable; or even when an input vector is multiplied by an optimizable matrix. The separate length scales are often used for interpretation, as a measure of relevance of each individual coordinate, which makes them a natural target for your study. Can your results be applied in this setting? Adding this would make the paper stronger.

(Minor) suggestions:
- In the abstract, perhaps you want to change “time series" into “time series forecasting” or “analysis of time series”, or something like that. Just “time series” doesn’t read like a “domain”.
- Lines 093-096, you mention non-identifiability of Matérn covariances, but you forget to mention that it only holds for dim <= 3.
- Line 132. “if for any” -> “if for all”.
- Line 198. “or they are supported on disjoint sets” - I strongly object to this intuition. Consider a non-degenerate Gaussian supported on $\mathbb{R}^2$ and a degenerate Gaussian supported on the line $\\{0\\} \times \mathbb{R} \subset \mathbb{R}^2$. These measures are orthogonal, but the support of the latter is a subset of the support of the former, the supports are not at all disjoint. Yes, you could exclude $\\{0\\} \times \mathbb{R}$ from the support of the former measure if you treat things up to probability 0 events, but this is not very natural and thus doesn't give a good intuition.
- Line 248. Please give a reference for the spectral density of Matérn kernels for the specific notion of the Fourier transform that you are using.
- Line 377. The notation $\\{ \pm s_1 \pm s_2 \pm \dots \pm s_m \\}$ is unclear, please expand on what you mean exactly by this.
- Lines 404-406. “In fact, even for simple kernels like the SE and Matérn kernels, whether the MLE is consistent remains open, and we still do not know whether the likelihood is unimodal or not.” - please support this claim by a reference.
- Please polish your reference list. As an example, in line 545, the word “gaussian” is missing a capital “G”.

---

> ### Author Response · Authors · 2024-11-18
>
> Thank you for the detailed comments and for acknowledging the novelty of our theoretical results. Below are our point-to-point responses, and we have revised the paper accordingly, with changes highlighted in orange.
>
> **General Combinations of Kernels**
>
> Thank you for this valuable question. We would like to clarify that Theorem 3.4 and Theorem 3.5, together with Theorem B.6 in the appendix, provides a general framework for determining identifiable parameters for any sum of products of stationary kernels holomorphic around 0. As concrete examples, we provided Theorems 3.6–3.9 (3.6 is new for the squared exponential ARD kernel, as you suggested, see the response below), which cover four distinct types of combinations.
>
> We realize that we did not adequately highlight Theorem B.6 in the main text. In the revised paper, we have explicitly referenced it and improved its writing to better reflect how it is used in conjunction with Theorem 3.4 and 3.5.
>
>
> **Extension to ARD RBF**
>
> Based on your insightful suggestion regarding ARD RBF kernels $K(x,x')=\sigma^2\exp(-\frac{1}{2}(x-x')^\top M(x-x'))$, we have extended this section to include a new theorem (3.6). This result demonstrates that all parameters, i.e., $\sigma^2$ and $M$, are identifiable. As a special case you pointed out, when $M$ is diagonal, $K(x,x')=\sigma^2\exp(-\frac{1}{2}\sum_{j=1}^p \frac{(x_j-x'_j)^2}{\ell_p})$, we proved that $\sigma^2,\ell_1,\cdots,\ell_p$ are all identifiable. This also serves as a great example to illustrate how to apply our Theorem 3.4 in practice. We greatly appreciate this suggestion, as it further highlights the generality of our theoretical framework.
>
> **Proofs**
>
> Thank you for your feedback. We agree that a concise overview of the key ideas in the main text is valuable. However, due to the technical nature and length of the proof, we chose to provide detailed exposition and full proofs in the appendices. This ensures that readers interested in the details can fully follow the arguments without overloading the main paper.
>
> To briefly summarize the proof of Theorem 3.4:
>
> 1. For stationary kernels holomorphic around $0$, we reduce the problem of equivalence of GP laws on $[0,T]^p$ to equivalence on $\mathbb{R}^p$.
>
> 2. Using the general criterion that GP laws are equivalent if and only if the difference of their kernels is a Hilbert-Schmidt operator, we reformulate the problem in terms of this operator property.
>
> 3. Finally, by leveraging the spectral isomorphism $Z(t) \leftrightarrow (\omega \mapsto e^{i\omega^T t})$, we translate the Hilbert-Schmidt condition into a spectral condition, which yields the desired results of Theorem 3.4.
>
> Regarding the term "holomorphic," we clarify: a kernel $K : \mathbb{R}^p \to \mathbb{R}$ is said to be holomorphic on a ball around $0$ in $\mathbb{C}^p$, if it has a unique holomorphic extension  $\tilde{K}$ to some ball $B\subset  \mathbb{C}^p$ around $0$, such that $\tilde{K}=K$ on $B \cap \mathbb{R}^p$. This standard definition bridges real-input kernels with complex-analytic methods. We added this clarification to the main text as well.
>
> We hope this clarification addresses your concerns while preserving the necessary focus on key ideas in the main text. We welcome further feedback on this point.
>
> **Empirical Studies**
>
> Thank you for this thoughtful comment. We acknowledge the limitations of our empirical results due to numerical challenges and sample size constraints. For instance, in our simulations with the RBF kernel, the covariance matrices have condition numbers on the order of $10^{-16}$ for sample sizes of 1300 or more, which explains the observed plateau in MLE variance. These limitations make it difficult to determine conclusively whether the MLEs are convergent.
>
> However, as we discussed in the second paragraph of Section 4 (Simulation), our primary focus is on step 1: the identifiability of kernel parameters. The subsequent steps—step 2, developing a theoretical understanding of the MLE, and step 3, analyzing the empirical behavior of the MLE—are indeed highly interesting but represent largely open problems. We explicitly highlight these as future directions in Section 5.
>
> Crucially, we did not make claims about the practical behavior of the MLE in this paper, as our theoretical framework and simulations were designed to address step 1: identifiability. We appreciate your feedback, which underscores the importance of these open problems and their potential for future exploration.
>
> **Writing** Thanks for the suggestion. We have carefully reviewed and revised the second paragraph of the Introduction.
>
> **Minor Comments**
> Thank you for your careful check and valuable comments. We have revised the paper accordingly.
>
> We thank the reviewer again for their detailed and constructive feedback, which has significantly improved the paper, and we remain open to further suggestions.

---

> > ### Comment · Reviewer_Ec9P · 2024-11-22
> >
> > I acknowledge reading the authors' rebuttal. It addressed most of my concerns. Therefore, I increase my score and recommend the acceptance of the paper.

---

### Official Review · Reviewer_ygPM · 2024-11-05

**Soundness:** 3
**Presentation:** 3
**Contribution:** 3
**Rating:** 6
**Confidence:** 2

**Summary:**

This paper investigates the identifiability of parameters in a Gaussian process (GP) model with a kernel that is stationary and holomorphic around zero. Identifiability in GPs is a critical issue for both parameter estimation and interpretability. However, according to the authors, identifiability remains insufficiently studied for a wide range of kernels, as most existing methods apply only to a limited set of kernels.

To address this, the authors introduce a new method for establishing the identifiability of GP parameters. Using this approach, they demonstrate that parameters for several well-known kernels are identifiable. They also derive conditions under which parameters of sums of products of kernels remain identifiable, extending the applicability of their method. Finally, experiments on datasets support their theoretical findings, confirming the identifiability of parameters across various kernels.

**Strengths:**

- The motivation for studying identifiability conditions for kernels of GP is strong, as it enables clearer interpretation of model parameters. Additionally, the focus on widely used kernels is a practical and relevant choice.

- The authors introduce a novel reasoning approach to derive the identifiability condition, which could be of interest to researchers beyond the immediate scope of this paper.

- Overall, the paper is well-written, and the authors have provided code to facilitate reproducibility of the results.

**Weaknesses:**

- While the results are presented for various kernels, the scope is still limited to stationary and holomorphic kernels around zero, which restricts the broader applicability of the identifiability condition.

- The experimental results are somewhat inconclusive. It would be helpful to clarify what should happen when parameters are identifiable. For instance, if the parameters are indeed non-identifiable, we might expect the estimated values to fluctuate across a range of possibilities. Can the experiment demonstrate that this is not occurring? Perhaps there is a statistical test that could be developed to verify whether the model is, in fact, identifiable. The observed reduction in variance is promising evidence, yet it leaves unanswered questions about cases where variance does not decrease. It would also be useful to test the behavior of parameters that are known to be non-identifiable—particularly those that would not be easily identifiable without the specific reasoning introduced in this paper.

**Questions:**

It is true that the decreasing variance with an increasing number of samples is reasonably convincing evidence that the parameters identified as "identifiable" are indeed identifiable. However, in cases where this pattern does not hold, do you have any additional arguments to support that the experiments still validate the theory's conclusions about identifiability? Conversely, what kind of behavior would you expect to see if a parameter were not identifiable? More broadly, is there a way to develop a meaningful statistical test to empirically confirm that the parameters theoretically classified as identifiable or non-identifiable actually exhibit these properties?

---

> ### Author Response · Authors · 2024-11-18
>
> Thank you for the overall positive score and for recognizing the novelty and significance of our work. Below, we provide a point-by-point response to address your questions and concerns, which helps us further improve the clarity and impact of our paper.
>
> **Broader Applicability**
>
> Thank you for the observation about the scope of our results. We would like to clarify the motivation and breadth of our contribution, as well as the challenges of extending identifiability results to non-stationary kernels.
>
> Within the class of stationary kernels, our work addresses a significant gap by covering a broad subset of widely used kernels, such as RBF, rational quadratic, periodic, and their additive and multiplicative combinations. These kernels are holomorphic around zero, a property we leveraged to establish their parameter identifiability using a novel theoretical framework. This includes cases that were not previously addressed by standard methods like the integral test, which was the main tool to study Matérn kernels.
>
> While we agree that non-stationary kernels present an interesting avenue for future research, they constitute a much larger and more diverse class of kernels, making a unified theoretical framework particularly challenging. For now, studying identifiability for non-stationary kernels remains an open problem, and we explicitly discuss this limitation in Section 5 of our paper.
>
> Importantly, the scope of our work aligns with prior studies, which often focus on a specific kernel class, such as Matérn kernels. For example, multiple papers have rigorously analyzed identifiability for Matérn kernels alone, given their practical relevance and mathematical tractability (e.g., Zhang, 2004; Anderes, 2010; Kaufman and Shaby, 2013). Similarly, we believe our focus on stationary holomorphic kernels represents a meaningful and impactful contribution.
>
> **Experimental Evidence**
>
> Thank you for your detailed feedback regarding the experimental results. We appreciate the opportunity to clarify the scope of our paper and the intent of the simulation section.
>
> As discussed in the second paragraph of Section 4, the broader study of GP parameter identifiability can be roughly divided into three main steps.
>
> 1. Determining which parameters are identifiable, which is the focus of our paper.
>
> Case 1.1: If a parameter is not identifiable, there **does not exist** any consistent estimator of the parameter, whether it is the MLE or another estimator.
>
> Case 1.2: If a parameter is identifiable, there **might exist** a consistent estimator for it. However, whether the MLE or another estimator is consistent is not guaranteed and remains an open problem, even for simple kernels like the squared exponential and Matérn kernels.
>
> 2. Establishing consistency of estimators: For identifiable parameters, determining whether a specific estimator (e.g., MLE or others) is consistent is an interesting and open problem that lies beyond the scope of this work.
>
> 3. Designing practical algorithms: Developing numerical methods to compute consistent estimators of identifiable parameters is another important but separate problem, also beyond the scope of our paper.
>
> Our work focuses solely on Step 1, introducing a theoretical framework to determine identifiability. Steps 2 and 3, while crucial, remain largely unexplored in the literature and are beyond the scope of this paper.
>
> The simulations in our paper (Section 4) are based on MLEs and serve primarily as a sanity check, as we do not provide any theoretical guarantees on MLE consistency. Exploring these guarantees is an important and challenging open problem, but, again, lies beyond the scope of this work.
>
> We appreciate the reviewer’s thoughtful suggestions regarding investigating variance patterns or designing a statistical test to empirically verify identifiability. While these are indeed interesting directions, they are inherently tied to the theoretical understanding of the consistency of specific estimators, which remains an open question. We highlight this as an important avenue for future research.
>
> We thank the reviewer for raising these insightful questions, which point to exciting directions for future research (see also Section 5). We remain open to further discussion or clarification.
>
> **Reference**
>
> H Zhang, Inconsistent estimation and asymptotically equal interpolations in model-based geostatistics, Journal of the American Statistical Association, 2004.
>
> E Anderes, On the consistent separation of scale and variance for Gaussian random fields, The Annals of Statistics, 2010.
>
> CG Kaufman, BA Shaby, The role of the range parameter for estimation and prediction in geostatistics, Biometrika, 2013.

---

> > ### Comment · Reviewer_ygPM · 2024-11-26
> > **Answer to rebutal**
> >
> > I thank the authors for their dedicated time and efforts in addressing my concerns. I acknowledge the inherent difficulty in developing a unified framework to handle non-stationary kernels. Regarding my concern about the experimental setup, while I understand the rationale for focusing primarily on determining identifiability, I believe it is essential to consistently validate the theoretical results presented in the paper, which forms the basis of my question (not about handling case 2 and 3 which I understand are not the focus of the paper).
> >
> > Although verifying the theoretical conclusions seems to rely on steps 2 and 3—areas that are not the main focus of the paper, I still recommend acceptance, recognizing the challenges of conducting such an evaluation rigorously.

---

### Official Review · Reviewer_wkyQ · 2024-11-06

**Soundness:** 4
**Presentation:** 4
**Contribution:** 4
**Rating:** 8
**Confidence:** 4

**Summary:**

This paper provides a new theoretical framework that can be used to determine identifiably of parameters in a stationary kernel under the infill asymptotic. Using this framework, the authors are able to determine the parameter identifiably of several popular kernels in the literature, extending the theoretical framework for the Matern class in the literature. Some numerical experiments are used to validate the theoretical findings.

**Strengths:**

This is a very well-written paper addressing an important research question in the literature. The theory is solid, especially Theorem 3.4, can be very useful for researchers in related areas.

**Weaknesses:**

No apparent weakness. (Note: I have reviewed this paper for another conference before, and all my previous questions have been addressed. Personally, I think this is a great work based on my research experience and should have been accepted.)

**Questions:**

I have to admit that I did not go through the proof line by line, but the arguments make intuitive sense.

---

> ### Author Response · Authors · 2024-11-18
>
> Thank you for your positive feedback and for recognizing the quality and significance of our work. Your previous comments have been invaluable in helping us improve the paper. Your statement that "this is a great work... and should have been accepted" this time is particularly encouraging.
>
> Although no specific questions were raised, we remain open to further clarification or discussion if needed. Given your strong endorsement of the paper’s contributions and its potential impact, we hope our work meets your expectations in every aspect, including a more positive score. In any case, we fully respect your decision and greatly appreciate your time and thoughtful review.

---

> > ### Comment · Reviewer_wkyQ · 2024-11-18
> > **Response to Response**
> >
> > I have raised my score.

---

### Author Response · Authors · 2024-12-01

We sincerely thank all reviewers for their helpful feedback, which has been invaluable in improving the manuscript. We are especially encouraged that all four reviewers gave positive scores, and we deeply appreciate the time and effort dedicated to evaluating our work.

---

### Meta-Review · Area_Chair_eejc · 2024-12-23

**Metareview:**

The paper investigates the problem of identifiability of the parameters of kernels which are holomorphic near zero.
In particular, the authors present a theoretical framework for studying the  identifiability of these kernels and apply it to
several kernels (squared exponential, periodic, rational quadratic, and cosine) and some of their sums and products, in Gaussian process (GP) regression.

Reviewers generally agree that the theoretical framework presented are novel and sound. On the other hand, the exposition, including the motivation for utilizing the framework of holomorphic functions from complex analysis, as well as empirical studies, should be improved.

**Additional Comments On Reviewer Discussion:**

- Reviewer: While the results are presented for various kernels, the scope is still limited to stationary and holomorphic kernels around zero, which restricts the broader applicability of the identifiability condition. The authors pointed out that non-stationary kernels form a  much broader class of kernels, making a unified theoretical framework more challenging.

- Reviewer Ec9P: "holomorphic" is a concept from complex analysis while the results are about kernels defined on $\mathbb{R}^n$. While the authors have defined this term in the revised version, I still find this link and motivation to be weak and should be improved.

- Reviewer Ec9P, iSg8: Empirical results do not illustrate the theoretical findings well. The authors pointed out that limitations of their empirical results are due to numerical challenges and sample size constraints.

The authors also pointed out that the scope of the paper is in determining which parameters are identifiable, not in developing consistent estimators for them, which is another separate and challenging direction.

With all of these points in mind, the current paper still makes an interesting and valuable contribution to the topic of GP parameter identifiability.

---

### Decision · Program_Chairs · 2025-01-22

Accept (Poster)